# De novo design of immunoglobulin-like domains

Tamuka M. Chidyausiku [1,2,3,8,10], Soraia R. Mendes [4,10], Jason C. Klima [1,2,9], Marta Nadal[5], Ulrich Eckhard [4], Jorge Roel-Touris[5], Scott Houliston[6,7], Tibisay Guevara[4], Hugh K. Haddox[2], Adam Moyer[2], Cheryl H. Arrowsmith [6,7], F. Xavier Gomis-Rüth [4] ✉, David Baker [1,2,3] ✉ & Enrique Marcos [5] ✉

Antibodies, and antibody derivatives such as nanobodies, contain immunoglobulin-like (Ig) β-sandwich scaffolds which anchor the hypervariable antigen-binding loops and constitute the largest growing class of drugs. Current engineering strategies for this class of compounds rely on naturally existing Ig frameworks, which can be hard to modify and have limitations in manufacturability, designability and range of action. Here, we develop design rules for the central feature of the Ig fold architecture—the non-local cross-β structure connecting the two β-sheets—and use these to design highly stable Ig domains de novo, confirm their structures through X-ray crystallography, and show they can correctly scaffold functional loops. Our approach opens the door to the design of antibody-like scaffolds with tailored structures and superior biophysical properties.

Immunoglobulin-like (Ig) domain scaffolds have two sandwiched β-sheets that are well-suited for anchoring antigen-binding hypervariable loops, as in antibodies and nanobodies. To date, approaches to engineering antibodies rely on naturally occurring Ig backbone frameworks, and mainly focus on optimizing the antigen-binding loops and/or multimeric formats for improving targeting efficiency or biophysical properties. Despite their exponential advance as protein therapeutics, engineered antibodies have significant limitations in terms of stability, manufacturing, size, and structure, among others. Several alternative antibody fragments, such as Fab (antigen-binding fragment) and scFv (single-chain variable fragment), and antibody-like scaffolds such as nanobodies have been engineered to address some of these limitations[1–3]. The β-sheet geometry in these antibody alternatives is kept very close to naturally existing Ig structures because it is much harder to modify the β-sheet structure than the variable loops.

De novo designing Ig domains with a wider range of core structures could expand the scope of antibody-engineering applications, but the design of β-sheet proteins remains a formidable challenge due to their structural irregularity and aggregation propensity[4]. Recent understanding of design rules controlling the curvature[5,6] and loop geometry in β-sheets[7,8] have enabled the design of β-barrels[6,9] and double-stranded β-helices[8], but the design principles for Ig domains and β-sandwiches, in general, are still poorly understood.

We set out to de novo design Ig fold structures, and began by considering the key aspects of the fold. The basic Ig domain[10,11] is a β-sandwich formed by 7-to-9 β-strands arranged in two antiparallel β-sheets facing each other, and connected through β-hairpins (within the same β-sheet) and β-arches[12] (crossovers between two opposing β-sheets). Natural Ig domains are structurally very diverse, often containing extra secondary structure elements and complex loop regions,

[1]Department of Biochemistry, University of Washington, Seattle, WA 98195, USA. [2]Institute for Protein Design, University of Washington, Seattle, WA 98195, USA. [3]Howard Hughes Medical Institute, University of Washington, Seattle, WA 98195, USA. [4]Proteolysis Laboratory, Department of Structural and Molecular Biology, Molecular Biology Institute of Barcelona (IBMB-CSIC), Baldiri Reixac 15, 08028 Barcelona, Spain. [5]Protein Design and Modeling Lab, Department of Structural and Molecular Biology, Molecular Biology Institute of Barcelona (IBMB-CSIC), Baldiri Reixac 15, 08028 Barcelona, Spain. [6]Structural Genomics Consortium, University of Toronto, Toronto, ON M5G 1L7, Canada. [7]Princess Margaret Cancer Centre and Department of Medical Biophysics, University of Toronto, Toronto, ON M5G 2M9, Canada. [8]Present address: Novartis Institutes for BioMedical Research Inc., San Diego, CA 92121, USA. [9]Present address: Encodia, Inc., San Diego, CA 92121, USA. [10]These authors contributed equally: Tamuka M. Chidyausiku, Soraia R. Mendes. ✉e-mail: xgrcri@ibmb.csic.es; dabaker@uw.edu; embcri@ibmb.csic.es

but they all share a protein core with a super-secondary structure "cross-β" motif that is common to most β-sandwiches: two antiparallel and interlocked β-arches[13] in which the first β-strands of each β-arch form one β-sheet, and the following β-strands cross and pair in the opposing β-sheet (Fig. 1). The four constituent cross-β strands (S$_2$, S$_3$, S$_5$, S$_6$) correspond to the B, C, E and F β-strands that build the common structural core of Ig domains found in nature[10,11], and for which some sequence signatures related to stability or function have been reported −e.g., a disulfide bridge between the B and F β-strands, a buried tryptophan in β-strand B[11,14] or the tyrosine corner[15] between β-strand C and the loop connecting β-strands E and F. The non-local cross-β structure (Fig. 1a) comprises two Greek key super-secondary structures[16,17] each involving four consecutive β-strands in which the first is paired to the last (Fig. 1b). Once the cross-β structures−which associate portions of the peptide chain distant along the linear sequence−are formed or designed, assembling the remainder of the peripheral β-strands is straightforward as it is only necessary to extend sequence-local β-hairpins out from the cross-β strands (Fig. 1b). Peripheral β-strands form later in the folding of Ig-like proteins[14,18], and are variable in number and structure across the different subtypes of Ig domains found in nature[10,11]. The cross-β motif also controls the overall β-sandwich geometry, which can be conveniently described by the rigid-body transformation parameters relating the two constituent β-

sheets−i.e., the distance and rotation along a vector connecting the two centers of the two opposing β-sheets, and the rotations around the two orthogonal vectors (Fig. 2a).

Here, we develop design principles controlling the cross-β motif structure of β-sandwiches. Based on these principles, we set up a computational approach for the de novo design of 7-stranded Ig domains with sequences and structures unexplored by natural ones. The structures of the designs were validated with X-ray crystallography, and for one of these, we show that it can correctly scaffold ligand-binding loops.

## Results

### Principles for designing cross-β motifs

We began by investigating how the structural requirements associated with cross-β motifs constrain the geometry of the two β-arches connecting the β-strands. Since β-arch connections have four possible sidechain orientation patterns[8] ("Out-Out", "Out-In", "In-Out" and "In-In") depending on whether the C$_\alpha$-C$_\beta$ vector of the β-strand residues preceding and following the β-arch connection point inwards ("In") or outwards ("Out") from the β-arch (Fig. 2b; Supplementary Fig. 1), there are sixteen possible cross-β motif connection orientations in total. For example, the "Out-Out/In-In" cross-β connection orientation means that the first and second β-arch connections have the "Out-Out" and

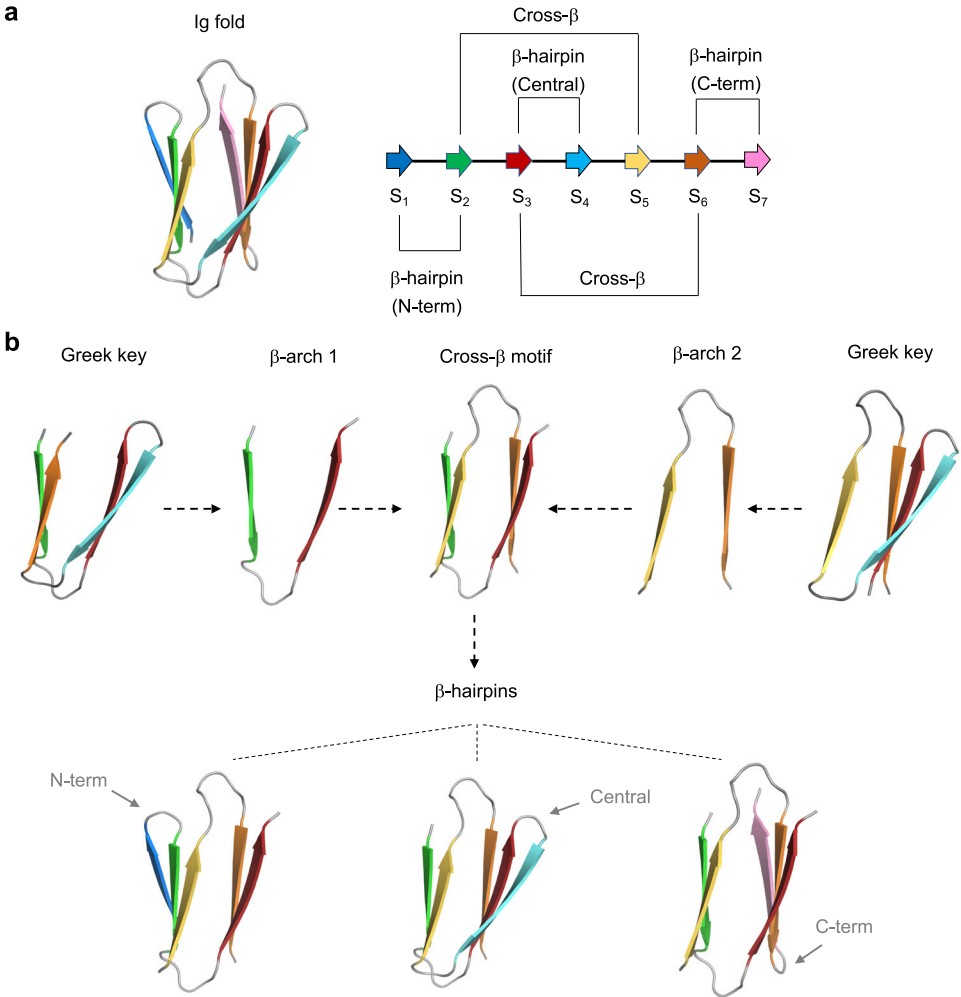

**Fig. 1 | Topology of immunoglobulin-like domains. a** Three-dimensional cartoon representation of an Ig structure formed by seven β-strands (left); and backbone hydrogen bond patterns (annotated thin lines) between paired β-strands along the sequence (right). Cross-β interactions have higher sequence separation (and higher contact order) than β-hairpins, which slows down folding. **b** β-arches of the cross-β motif belong to two contiguous and distinct Greek key motifs: with 2 β-strands in each β-sheet (left); and with 3 β-strands in one β-sheet and 1 β-strand in the other (right). From the folding and design perspective, the main limiting factor for correctly assembling the Ig structure is formation of the cross-β motif, since the three β-hairpins can form independently of one another.

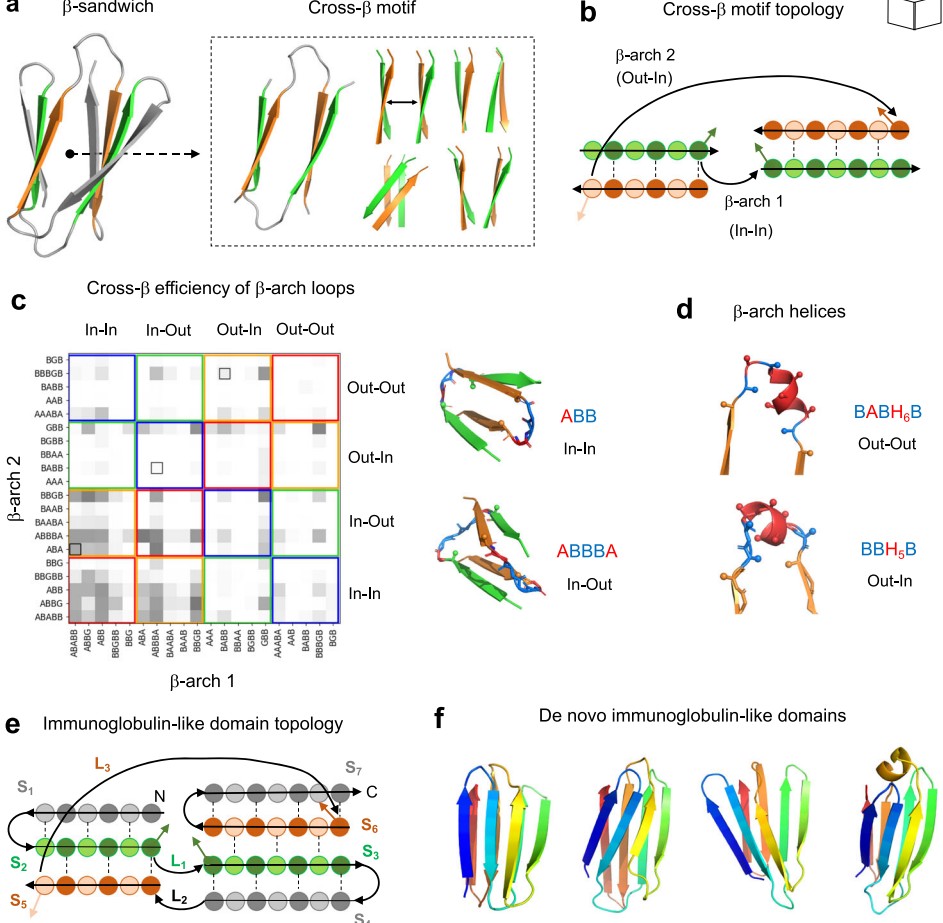

**Fig. 2 | Design rules for cross-β motifs in β-sandwiches. a** Cartoon representation of a 7-stranded immunoglobulin-like domain model formed by two β-sheets packing face-to-face, and the corresponding cross-β motif, which generates translations and rotations between the two opposing β-sheets. **b** Topology diagram of a cross-β motif with circles and arrows representing β-strand residue positions and connections, respectively. Dark- and light-colored circles correspond to residues with sidechains pointing inwards or outwards from the β-sandwich, respectively. **c** Efficiency of pairs of common β-arch loop geometries (described with ABEGO backbone torsions) in forming cross-β motifs obtained from Rosetta folding simulations (gray shaded squares). Loop geometries were classified in four groups according to the sidechain directions of the adjacent residues. Colored squares group pairs of loops that, due to their sidechain orientations, have different requirements in β-strand length: in red or blue, if all β-strands need an odd or even

number of residues, respectively; in green, if the β-strands of the first and second sheet need an odd and even number of residues, respectively; and in yellow for the opposite case (even and odd number of residues for the first and second sheet, respectively). Black-outlined boxes highlight loop combinations observed in natural Ig domains. On the right, examples of changes in cross-β motif geometry linked to β-arch loop geometry. **d** β-arch helices are formed by a short α-helix connected to the adjacent β-strands with short loops, and are complementary to β-arch loops for connecting cross-β motifs. **e** Topology diagram of a 7-stranded Ig domain. β-strands and β-arch loops are indicated as $S_i$ and $L_i$, respectively, where i is the corresponding number. **f** Examples of de novo designed Ig backbones generated with different geometries and β-arch connections following the described rules, colored from N-terminus (blue) to C-terminus (red).

"In-In" orientations, respectively. Due to the alternating pleating of β-strands, the cross-β connection orientation and the length of the β-strands in the two β-sheets are strongly coupled: if paired β-strands have no register shift, they must be odd-numbered in four of the possible cross-β orientations, even-numbered in four of the other possible cross-β orientations, and odd-numbered in one of the two β-sheets and even-numbered in the other β-sheet in the remaining eight cases. Guided by this principle, we studied the efficiency in forming cross-β motifs of highly structured β-arch connections; too flexible β-arches can hinder folding as they increase the protein contact order[19]—the average sequence separation between contacting residues—which slows down folding. The cross-β motif is the highest contact order part of the Ig fold architecture, and thus the rate of formation of this structure likely determines the overall rate of folding and thus contributes to the balance between folding and aggregation; once the cross-β motif is formed, folding is likely completed rapidly as the remaining β-hairpins are sequence-local (Fig. 1b).

We generated cross-β motifs exploring combinations of short β-arch loops frequently observed in naturally occurring proteins and spanning the sixteen possible sidechain orientations (Supplementary Fig. 1), along with β-strand length, using Rosetta folding simulations with a sequence-independent model[7,20] biased by the ABEGO torsion bins specifying desired loop geometries[21] (Fig. 2c). It is convenient to describe the backbone geometry of loop residue positions with ABEGO torsion bins representing different areas of the Ramachandran plot ("A", right-handed α-helix region; "B", extended region; "E", extended region with positive φ; "G", left-handed α-helix region; and "O", if the peptide bond deviates from planarity) (see Supplementary Fig. 1a for a definition). For cross-β motifs to form, the geometry of the two β-arch loops must allow the concerted spanning of the proper distance along the β-sheet pairing direction and along an axis connecting the two opposing β-sheets so that the two following β-strands cross and switch the order of β-strand pairing in the opposite β-sheet (Supplementary Fig. 2). Multiple pairs of β-arch loops with the same or different ABEGO

torsion bins were found to fulfill these geometrical requirements (Fig. 2c), with sampled ranges of cross-β geometrical parameter values similar to or broader than those found in naturally occurring Ig domains (Supplementary Fig. 3). For example, β-arch loops "ABB" and "ABBBA" strongly favor cross-β motifs but with twist rotations (Supplementary Fig. 4) in opposite directions (Fig. 2c, right). Of the short β-arch loops we considered for design, only a few are present in the cross-β motifs of naturally occurring Ig domains (Fig. 2c), which are mostly built by longer or hypervariable loops (as is the case of the first β-arch). We next explored the efficiency of short α-helices (spanning 4–6 residues) connecting the two β-strands through short loops (of 1–3 residues) which we refer to as "β-arch helices". For cross-β motifs formed with β-arch helices, we identified efficient loop-helix-loop patterns (i.e., helix length together with adjacent loop ABEGO-types) for the four possible β-arch sidechain orientations (Supplementary Fig. 5). Overall, the formation and structure of cross-β motifs can in this way be encoded by combining β-arch loops and/or β-arch helices of specific geometry with β-strands compatible in terms of length and sidechain orientations.

## Computational design of Ig domains

Based on these rules relating β-arch connections with cross-β motifs, we de novo designed 7-stranded Ig topologies (Fig. 2e, f). We generated protein backbones by Rosetta Monte Carlo fragment assembly using blueprints[7,20] specifying secondary structures and ABEGO torsion bins, together with hydrogen bond constraints specifying β-strand pairing. We explored combinations of β-strand lengths (between 5 and 8 residues) and register shifts between paired β-strands 3 and 6 (between 0 and 2 residues). β-arches 1 and 3 are those involved in the cross-β motif, and their connections were built with loop ABEGO-types having high cross-β propensity, as described above. We reasoned that β-arch helices may fit better in β-arch 3 than in β-arch 1 (Fig. 2e), which by construction is more embedded in the core, and explored topology combinations combining β-arch 1 loops with β-arch 3 helices. The three β-hairpin loops were designed with two residues for proper control of the orientation between the two paired β-strands according to the ββ-rule[7]. Those topology combinations with β-strand lengths incompatible with the expected sidechain orientations of each β-arch and β-hairpin connection were automatically discarded. We then carried out Rosetta sequence design calculations[22,23] for the generated backbones. Loops were designed using consensus sequence profiles derived from fragments with the same ABEGO backbone torsions. Cysteines were not allowed during design to avoid dependence of correct folding on disulfide bond formation (in contrast to most natural Ig domains). As an implicit negative design strategy against edge-to-edge interactions promoting aggregation, we incorporated at least one inward-facing polar or charged amino acid (TQKRE)[24] into each solvent-exposed edge β-strand. Sequences were ranked based on energy and sidechain packing metrics, as well as local sequence-structure compatibility assessed by 9-mer fragment quality analysis[4]. Folding of the top-ranked designs was quickly screened by biased forward folding simulations[5], and those with near-native sampling were subjected to Rosetta ab initio folding simulations from the extended chain[25]. The extent to which the designed sequences encode the designed structures was also assessed through AlphaFold[26] or RoseTTAFold[27] structure prediction calculations (see below).

## Biochemical characterization of the designs

For experimental characterization, we selected 31 designs predicted to fold correctly by ab initio structure prediction (Fig. 3a, b); 29 of which had AlphaFold or RoseTTAFold predicted models with pLDDT > 80 and $C_{\alpha}$ atom root mean square deviations (Cα-RMSDs) <2 Å to the design models (Supplementary Table 1). The designed sequences contain between 66 and 79 amino acids and are unrelated to naturally occurring sequences, with Blast[28] (E-values >0.1) and more sensitive

sequence-profile searches[29,30] finding very weak or no remote homology (E-values >0.003) (Supplementary Table 2). The designs also differ substantially from natural Ig domains in global structure (with an average ± s.d. TM-score[31] of 0.54 ± 0.06; Supplementary Fig. 6), and cross-β twist rotation (close to zero, which are infrequent in natural Ig domains; Supplementary Table 3). We obtained synthetic genes encoding for the designed amino acid sequences (design names are dIGn, where "dIG" stands for "designed ImmunoGlobulin" and "n" is the design number). We expressed them in *Escherichia coli,* and purified them by affinity and size-exclusion chromatography. Overall, 24 designs were present in the soluble fraction and 8 were monodisperse, had far-UV circular dichroism spectra compatible with an all-β protein structure, and were thermostable ($T_m > 95\,°C$, except for dIG21 with $T_m > 75\,°C$) (Fig. 3c, Supplementary Table 4, Supplementary Figs. 7 and 8). In size-exclusion chromatography combined with multi-angle light scattering (SEC-MALS), five designs were dimeric, one was monomeric (dIG21) and another one (dIG8) was found in equilibrium between monomer and dimer (Fig. 4a, Supplementary Figs. 7, 8 and 9). The monomeric design had a well-dispersed $^1H$-$^{15}N$ HSQC nuclear magnetic resonance (NMR) spectrum consistent with a well-folded β-sheet structure (Supplementary Fig. 10).

## Structural characterization of a dimeric de novo Ig design

The most stable design, dIG14, remained folded at 5 M guanidine hydrochloride (GdnCl) (Fig. 3d), had a well-dispersed $^1H$-$^{15}N$ HSQC spectra (Supplementary Fig. 10) and was found to be dimeric by SEC-MALS (Fig. 4a). To gain structural insight on its dimerization mechanism, we solved a crystal structure at 2.4 Å resolution (Fig. 4b, c, Supplementary Table 5) and found it was in excellent agreement with the computational model over the first five β-strands and their connections (Cα-RMSD of 0.8 Å; Fig. 4c). By contrast, the C-terminal region had three main differences: β-arch 3 helix was found in a different orientation, the register between paired β-strands 6 and 3 shifted by two β-strand positions (Fig. 4c, right inset), and the C-terminal β-strand flipped out of the structure, being disordered. This conformational difference altered the cross-β structure, exposed the protein core and formed an edge-to-edge dimer interface mediated by two antiparallel β-strand pairs (between β-strands 1 and 6 of each protomer), overall forming a 12-stranded β-sandwich (Fig. 4b). AlphaFold and RoseTTAFold predictions recapitulated the design model and did not predict these conformational differences, but the pLDDT values in the β-arch helix were quite low compared with the rest of the structure (Fig. 4d; Supplementary Fig. 11). Rosetta ab initio folding simulations sampled conformations closer to the crystal structure with energies similar to the design (Supplementary Fig. 11). Structure prediction of dIG14 as a homodimer with AlphaFold-Multimer[32] generated models closer to the crystal structure (Fig. 4d) despite formation of an incorrect dimer interface (Supplementary Fig. 11); the conformational differences between the design and crystal structure may be driven at least in part by the energetics of dimer interface formation.

## Structural characterization and functionalization of a monomeric de novo designed Ig scaffold

For the dIG8 design, crystallization trials yielded no hits, but we reasoned that a disulfide bond could further rigidify the structure and promote crystallization. As disulfide bonds with high sequence separation are more stabilizing due to greater unfolded state entropy reduction, we computationally designed disulfide bonds between β-strands not forming a β-hairpin using a hash-based disulfide placement protocol[33] which searches for transformations between pairs of residue positions compatible with naturally occurring disulfide bond geometries (see "Methods"). We designed the double mutant dIG8-CC (V21C, V60C) (Fig. 5a), which, like the parental protein (Supplementary Fig. 7), was well-expressed, thermostable and was found in an equilibrium between monomers and dimers by SEC-MALS (Fig. 5b). We were

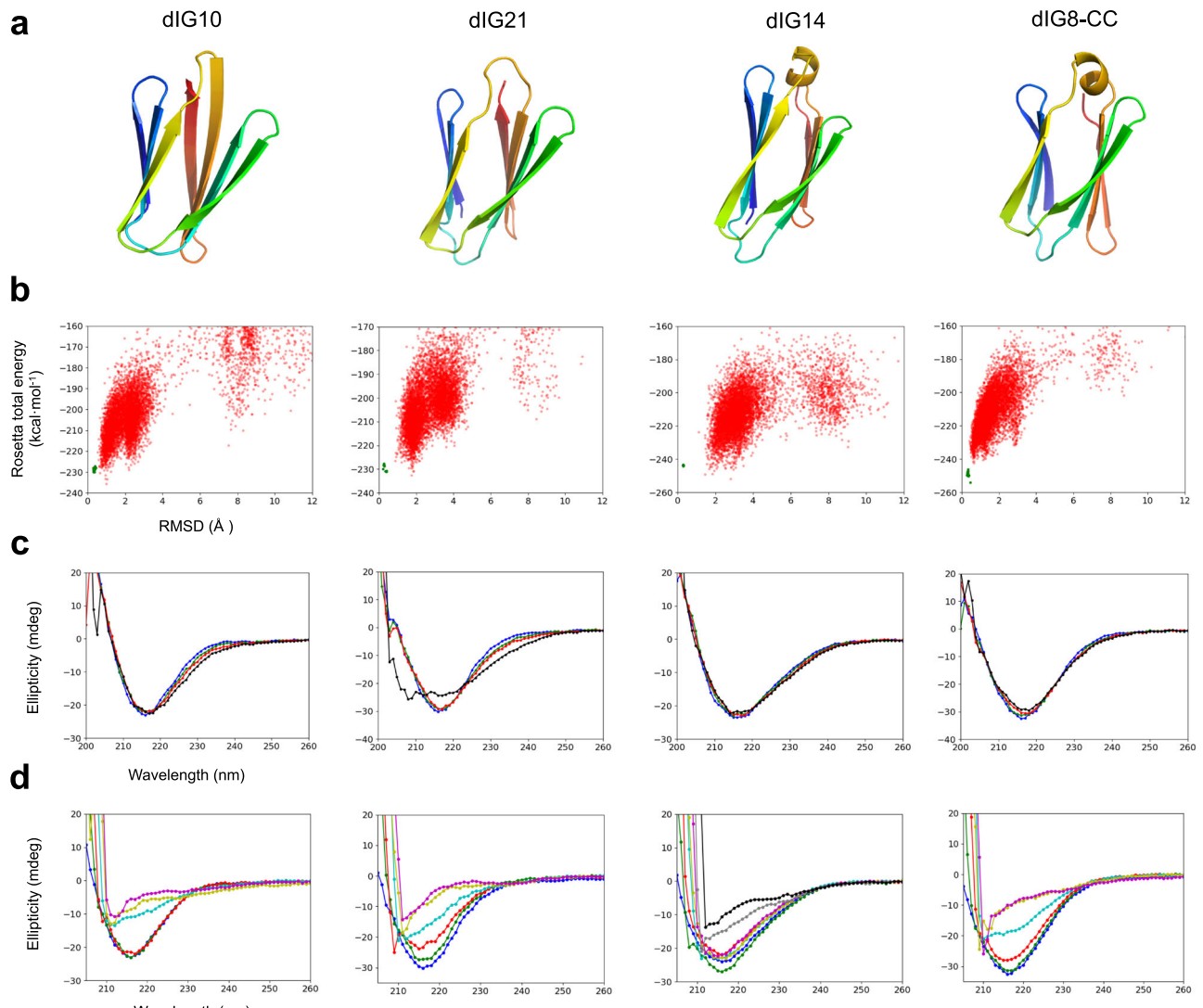

**Fig. 3 | Folding and stability of designed proteins. a** Examples of design models. **b** Simulated folding energy landscapes, with each dot representing the lowest energy structure obtained from ab initio folding trajectories starting from an extended chain (red dots) or local relaxation of the designed structure (green dots). The x-axis depicts the Cα-RMSD from the designed model and the y-axis, the Rosetta all-atom energy. **c** Far-ultraviolet circular dichroism spectra (blue: 25 °C; green: 55 °C; red: 75 °C; black: 95 °C). **d** Far-ultraviolet circular dichroism spectra at different guanidine hydrochloride concentrations and at 25 °C (blue: 0 M; green: 1 M; red: 2 M; cyan: 3 M; yellow: 4 M; magenta: 5 M; gray: 6 M; black: 7 M).

able to obtain two crystal structures of dIG8-CC in two different space groups, with data to 2.05 and 2.30 Å resolution by molecular replacement using the design and RoseTTAFold predicted models (Supplementary Table 5). The asymmetric unit of both crystal structures contained four protomers, and all of them closely matched the computational model with Cα-RMSDs ranging between 1.0 and 1.3 Å (Fig. 5c). The designed cross-β motif combines a β-arch loop (ABABB) with a β-arch helix (BB-H5-B), and both were well recapitulated (Cα-RMSDs ranging between 0.7 and 1.0 Å for the two connections) across the eight monomer copies, suggesting high structural preorganization of the designed connections (Fig. 5d). The sidechain of residue C21 was found in two different conformations, disulfide-bonded with C60 as in the design and unbound (Supplementary Table 6), which suggests low stability of the disulfide bond (Supplementary Fig. 12) and that it is not essential for proper folding of dIG8-CC. This is consistent with the high stability determined for parental dIG8 without the disulfide bridge (Supplementary Fig. 7).

The crystal structures also revealed an edge-to-edge dimer interface between the N- and C-terminal β-strands, overall forming a

14-stranded β-sandwich (Fig. 5e). Docking calculations on dIG8-CC suggested that the β-sandwich edge formed by the two terminal β-strands is more dimerization-prone than the opposite edge (Supplementary Fig. 13), mainly due to a more symmetrical backbone arrangement and complementary hydrophobic and salt-bridge interactions in the former, and the presence of more inward-pointing charged residues in the latter. In contrast to dIG14, Alpha-Fold correctly predicted the dIG8-CC monomer crystal structure with very high confidence across all residues and did not change that prediction in the context of the homodimer. The closest Ig structural analogs found across the PDB and the AlphaFold Protein Structure Database[34] had a TM-score ≤0.65 (Supplementary Fig. 14); and contained more irregular β-strands, longer loops, and differences in the β-strand pairing organization.

We next sought to investigate whether de novo designed immunoglobulins could be functionalized by scaffolding ligand-binding loops. We set out to computationally graft an EF-hand calcium-binding motif (PDB accession code 1NKF) into the β-hairpins of dIG8-CC. To facilitate motif grafting, we designed N-terminal linkers containing

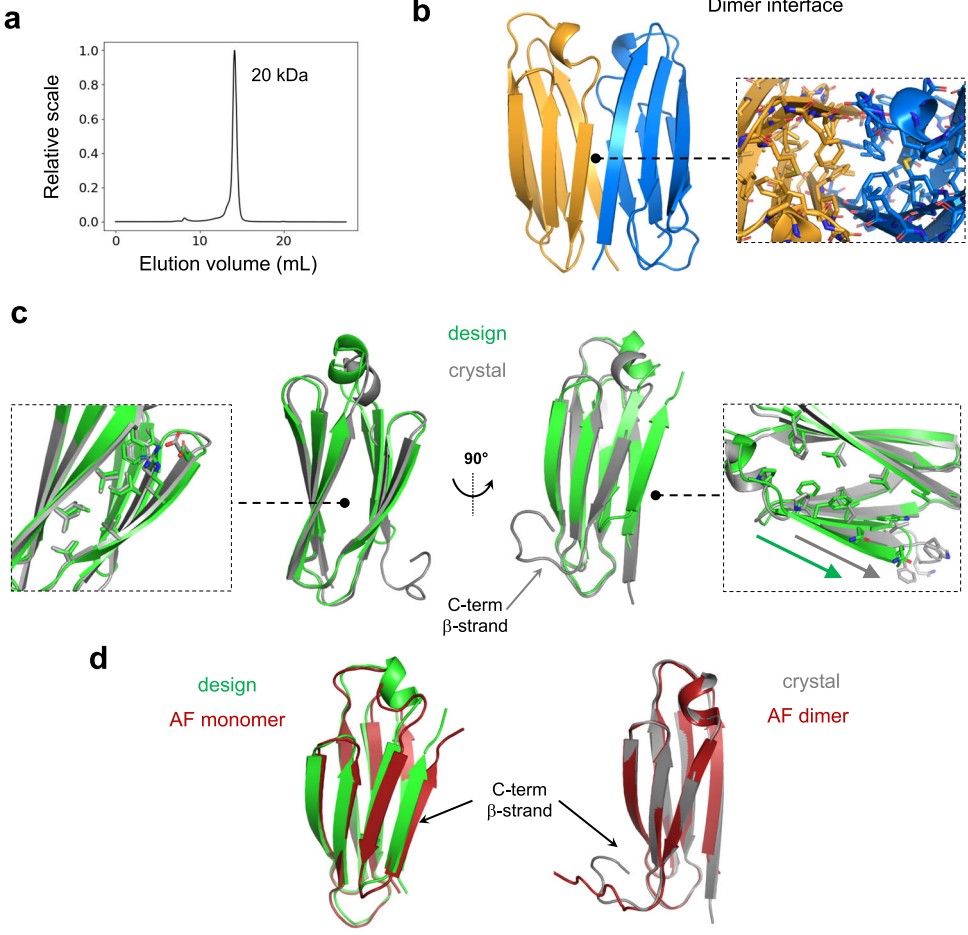

**Fig. 4 | Crystal structure of the dIG14 dimer. a** SEC-MALS analysis of dIG14 estimates a molecular weight corresponding to a dimer ($M_w$ monomer = 9.7 kDa). **b** Crystal structure of the homodimer interface formed by antiparallel pairing between β-strands 1 and 6 enabled by flipping out of the C-terminal β-strand; the monomer core becomes more accessible and the interface is primarily formed by hydrophobic contacts (right inset). PDB accession code of the dIG14 crystal structure: 7SKP. **c** dIG14 design model (green) in comparison with the crystal structure (gray, chain B). Sidechain packing interactions in the non-terminal edge β-strands were well recapitulated in the crystal structure (left inset). A shift in β-strand pairing register observed in the crystal structure is highlighted by the two colored arrows (right inset). **d** The AlphaFold monomer prediction (left) superimposes well with the design model (Cα-RMSD 1.0 Å); while AlphaFold-Multimer (right) correctly predicts the monomer subunits in the crystal structure (Cα-RMSD 0.6 Å, except for the C-terminal disordered β-strand).

---

between 0 and 3 residues with an extended backbone conformation, and C-terminal linkers containing between 0 and 10 residues keeping the α-helical secondary structure of the C-terminal side of the EF-hand motif. We selected 12 designs for experimental testing with minimal linker lengths and spanning the three insertion sites. Design EF61_dIG8-CC (Fig. 5f), with the EF-hand motif grafted at the C-terminal β-hairpin of dIG8-CC after residue 61, was the best expressed and monodisperse by size-exclusion chromatography, and was found to be thermostable by far-UV circular dichroism (Fig. 5g), as was the parent design dIG8-CC. Since EF-hand motifs generally bind terbium, we assessed ligand-binding by terbium luminescence, which can be sensitized by energy transfer[35] from a proximal tyrosine residue on the grafted EF-hand motif upon excitation at 280 nm wavelength. For increasing luminescence signal-to-noise ratio, we carried out time-resolved luminescence measurements taking advantage of the long luminescence lifetime of terbium[36,37]. EF61_dIG8-CC mixed with 100 μM TbCl₃ displayed a 10-fold higher luminescence emission intensity at 544 nm than dIG8-CC without the EF-hand motif (Fig. 5h). Tb³⁺ titrations in the presence of EF61_dIG8-CC displayed a hyperbolic increase in luminescence with increasing Tb³⁺ concentrations (Fig. 5i; Supplementary Fig. 15a). In competitive binding titrations, Tb³⁺ luminescence intensity decreased with increasing Ca²⁺ concentrations, showing that Ca²⁺ competes with Tb³⁺ for the grafted EF-hand motif (Supplementary Fig. 15b).

## Discussion

Since initial attempts in the early 90's[38–40], the de novo design of globular β-sheet proteins with high-resolution structural validation had remained elusive until very recently, when they were enabled by considerable advances in our understanding of how to program the curvature of β-sheets and the orientation of their connecting loops into an amino-acid sequence. Here, we describe the successful de novo design of an immunoglobulin-like domain with high stability and accuracy, which had not been achieved yet and was confirmed by crystal structures. This success became possible by elucidating the requirements for effective formation of cross-β motifs, which establish the non-local central core of Ig folds by structuring β-arch connections through short loops and helices, while favoring sidechain orientations compatible with the length and pleating of the sandwiched β-sheets.

The cross-β motifs of our designs differ from natural ones in several ways. Our cross-β motifs are formed by combining short β-arch loops not seen in natural Ig domains (Fig. 2c), which generally have more complex loops (including a complementarity-determining region (CDR) in the first β-arch of the cross-β motif found in antigen-binding regions of antibodies), and are stabilized by hydrophobic interactions without incorporating sequence motifs typically found in the core strands B, C, E, and F of natural Ig domains. For example, the disulfide bond of dIG8-CC is between two β-strands paired in the same

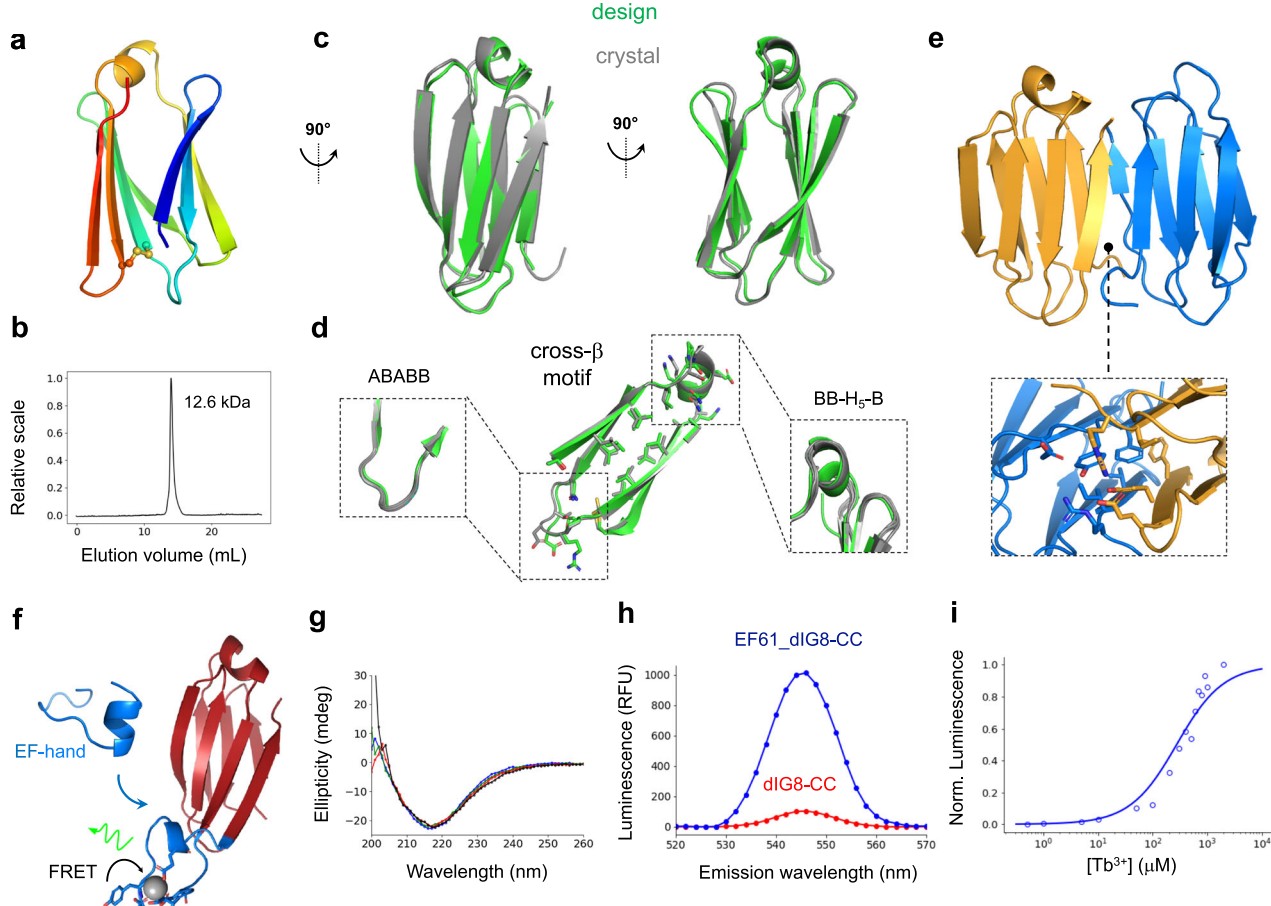

**Fig. 5 | Crystal structure of dIG8-CC and functional loop scaffolding. a** Design model of dIG8-CC with a disulfide bridge (spheres) between β-strands 3 and 6. **b** SEC-MALS analysis of dIG8-CC estimates a molecular weight between monomer (8.3 kDa) and dimer (16.6 kDa). **c** Design model (green) in comparison with the crystal structure with PDB accession code 7SKP (gray, chain C). **d** Cross-β motif connections and core sidechain interactions in the design and the crystal structure. The β-arch helix and loop conformations are well preserved across monomer copies in the crystal asymmetric units (insets). **e** Crystal homodimer interface by parallel pairing between the two terminal β-strands, which are stabilized through hydrophobic and salt-bridge interactions (inset). **f** Computational model of dIG8-CC with a grafted EF-hand motif (design EF61_dIG8-CC, cartoon), showing $Tb^{3+}$ (sphere) bound to EF-hand motif residues (sticks). $Tb^{3+}$ luminescence is sensitized by absorption of light (purple) by a proximal tyrosine residue on the EF-hand motif with subsequent fluorescence resonance energy transfer (FRET) to $Tb^{3+}$, resulting in $Tb^{3+}$ luminescence (green). **g** Far-ultraviolet circular dichroism spectra of EF61_dIG8-CC without $Tb^{3+}$ (blue: 25 °C; green: 55 °C; red: 75 °C; black: 95 °C). **h** Time-resolved luminescence emission spectra in 100 μM $Tb^{3+}$ final concentrations for EF61_dIG8-CC (blue) and dIG8-CC (red) at 20 μM. Time-resolved luminescence intensity is given in relative fluorescence units (RFU). **i** $Tb^{3+}$ concentration-dependent time-resolved luminescence intensity of 20 μM EF61_dIG8-CC using excitation wavelength $\lambda_{ex} = 280$ nm and emission wavelength $\lambda_{em} = 544$ nm. Normalized intensities are fit to a one-site binding model by nonlinear least squares regression ($K_d = 267$ μM).

β-sheet in contrast to the sheet-to-sheet disulfide bridge found between strands B and F in many Ig domains. The tyrosine corner which stabilizes Greek keys in many natural β-barrels and β-sandwiches[15,18] was also not needed in our designs. These differences in sequence requirements reflect the substantial structural differences between our designs and natural Ig domains. The designs contain cross-β motifs less twisted than those from natural Ig domains, and their overall structural (average TM-score of 0.54) and sequence (Supplementary Table 2) similarity is very low (HHPred did identify matches to short segments of β-sandwiches, including one Ig domain (PDB accession code 2R39), with locally similar alternating patterns of hydrophobic and polar amino acids typical of β-strands).

Several of the designs tended to dimerize in solution, highlighting design challenges in preventing self-interactions between β-sheets. Solvent-exposed β-strand edges favor intermolecular β-strand pairing through backbone hydrogen bonds (between the unpaired NH- and CO- groups) and hydrophobic interactions at the interface between monomers. As in previous de novo β-sheet design

studies[5,7,8], we used an implicit negative design strategy to disfavor association by favoring polar or charged amino acids at inward-facing positions of the edge β-strands to weaken interface sidechain interactions. Explicit negative design against possible edge-to-edge dimer interfaces is an alternative, but remains challenging as it requires enumerating many possible negative states: the crystal structures of two designs show two possible interfaces (one including structural rearrangement of the monomer), and we cannot rule out the possibility that other dimer interfaces formed in designs that were not crystallized (via parallel or antiparallel edge-strand pairing with varied register shifts). Alternatively, negative design against edge-to-edge interfaces can be encoded in protein backbone irregularities−e.g., β-bulges, prolines or short protective β-strands−disfavoring the ideal geometry for hydrogen-bonded β-strand pairing[41].

The edge-to-edge dimer interfaces in the crystal structures of our designs differ from those found between the heavy- and light-chains of antibodies, which are arranged face-to-face. For engineering antibody-

like formats presenting several loops targeting one or multiple epitopes, designing dimeric Ig interfaces through the β-sandwich edge formed by the terminal β-strands has the advantage over face-to-face dimers of decreasing the number of exposed β-strand edges, thereby reducing aggregation-propensity. It will likely be useful to custom-design both edge-to-edge and face-to-face dimers from our de novo Ig domains; these would present loops from the two monomers in different relative orientations, and depending on the target structure and the loops involved, one of these two arrangements will likely be better suited than the other for designing shape-complementary binding interfaces. Another advantage of controlling the orientation of dimer interfaces is that the N- and C-termini of the two monomeric subunits can be positioned in close proximity to allow fusion through short or compact connections into rigid and hyperstable single-chain constructs—similar in spirit to single-chain variable fragments (scFvs) but with greater structural control and higher stability.

The high stability of our designs opens up exciting possibilities for grafting functional loops, as shown for the EF-hand terbium-binding motif inserted into the C-terminal β-hairpin of dIG8-CC. The β-hairpins in our scaffolds can be readily extended to incorporate ligand- and protein-binding motifs, functional peptide motifs, or complementarity-determining regions (CDRs) of antibodies or nanobodies (it is likely more straightforward to insert functional loops into β-hairpins than into β-arches, since the latter tend to form more slowly and need to be highly structured, but this remains to be studied and may vary depending on the loop to be inserted). In antibodies, the CDRs are located on one side of the β-sandwich (at the bottom given the orientation displayed in Figs. 1–5), and we inserted the terbium-binding motif on this side, but the robustness of our scaffolds could allow insertions on the other side as well. Ultimately, achieving the structural control over the Ig backbone together with the high expression levels and stability of de novo designed proteins in general should lead to a versatile generation of antibody-like scaffolds with improved properties.

## Methods

### Structural analysis of β-arch loops

β-arch loops of <9 residues were collected from a non-redundant set of 5857 PDB structures with sequence identity <30% and resolution ≤2.0 Å. They were identified by first assigning the secondary structure with DSSP[42], and ensuring they were connecting β-strands with no hydrogen-bond pairing between them (the first and last residue of each assigned β-strand were considered the end residues connecting to the loops). The ABEGO torsion bins of each loop position was assigned based on their φ/ψ backbone dihedrals as defined in Supplementary Fig. 1a. The sidechain orientations of the two residues (i and j) preceding and following the β-arch loop are a function of the relative orientation between their $C_\alpha$-$C_\beta$ vector and the translation vector ($v_1$) connecting their $C_\alpha$ atoms, as shown in Supplementary Fig. 1b. The β-arch sliding distance was calculated as the dot product between $v_1$ and the $CO$ vector of the preceding residue ($v_1 \cdot CO_i$), which points along the β-sheet hydrogen bond direction. If the dot product between $v_1$ and the $C_\alpha$-$C_\beta$ vector of the preceding residue is negative, then the sliding distance is calculated as $v_1 \cdot -CO_i$. The β-arch twist was calculated as the dihedral between positions $C_\alpha$ (i-2), $C_\alpha$ (i), $C_\alpha$ (j), and $C_\alpha$ (j + 2).

### Cross-β motif analysis

To extract the cross-β geometrical parameters we calculated the rigid body transformation between two reference frames defined at the two β-sheets comprising the cross-β motif. For the first β-sheet (formed by the two N-terminal strands, 1 and 3, of the motif), the reference frame was built with the vectors $S_1$, which defines the direction of β-strand 1 (from N to C-termini), $S_{31}$, which connects the centers of the two strands (Supplementary Fig. 2), and $P_N$ as the vector orthogonal to the

β-sheet calculated as the cross product between the $S_1$ and $S_{31}$ vectors ($P_N = S_1 \times S_{31}$). For the second β-sheet (formed by the two C-terminal strands, 2 and 4, of the motif), the reference frame was calculated in the same way with the equivalent vectors $S_4$, $S_{24}$, and $P_C$. To minimize the dependence of cross-β parameters on differences in the internal geometry of β-strands from the two different β-sheets, we pre-generated a template antiparallel strand dimer that, before calculating the transform, is superimposed on each of the two strand dimers of the cross-β motif. The transform rotational angles were calculated as the Euler angles of the transform (twist, roll, and tilt). The cross-β motif distance was calculated between the centers of the two strand dimers. The β-arch sliding distance in a cross-β motif was calculated as the dot product between the translation vectors and the vector $S_{31}$.

### Structural analysis of naturally occurring immunoglobulin-like domains

We searched for Ig-like domains classified in SCOP[43] as "Ig-like beta-sandwich" folds (SCOP ID 2000051) and selected those with X-ray resolution ≤2.5 Å, yielding a total of 467 annotated domains.

### Protein backbone generation and sequence design

We specified blueprint files for each target protein topology and constructed poly-valine backbones with the RosettaScripts[44] implementation of the Blueprint Builder[7] mover, which carries out Monte Carlo fragment assembly using 9- and 3-residue fragments picked based on the secondary structure and ABEGO torsion bins specified at each residue position. We used the fldsgn_cen centroid scoring function with reweighted terms accounting for backbone hydrogen bonding (lr_hb_bb) and planarity of the peptide bond (omega).

For constructing cross-β motifs, we followed a two-step procedure. First, the two N-terminal strands of the motif (strands 1 and 3) were generated as antiparallel β-strand dimers of desired length from φ/ψ values typical of β-strands (extended region of the Ramachandran plot) and relaxed using hydrogen-bond pairing restraints. Second, the cross-β loops and C-terminal strands (strands 2 and 4) were then appended by fragment assembly using the Blueprint Builder, as described above, combined with a strand pairing energy bonus between strands 2 and 4. We assign the two N-terminal strands to different chains (A and B), and the resulting jump between the two chains allows to fold the two C-terminal strands independent of each other. Then, the secondary structures of the resulting backbones were calculated by DSSP[42] and those with a secondary structure identity to that defined in the blueprints below 90% were discarded to guarantee correct strand pairing formation. The filtered backbones needed to fulfill two additional properties to be considered a cross-β motif: (1) the two C-terminal strands must form antiparallel strand pairing with each other, but not with any of the N-terminal strands (to guarantee β-sandwich formation); (2) the two β-arches must cross. For the latter, we checked crossing based on the relative orientation between the two vectors orthogonal to each of the two β-sheet planes packing face-to-face. The $P_N$ vector orthogonal to the β-sheet formed by the two N-terminal strands is calculated as the cross product between the $S_1$ and $S_{31}$ vectors ($P_N = S_1 \times S_{31}$) as described above. The $P_C$ vector orthogonal to the β-sheet formed by C-terminal strands is calculated similarly as $P_C = S_4 \times S_{24}$ as described above. If the two orthogonal vectors are parallel (if $P_N \cdot P_C > 0$) the two β-arches were considered to cross.

For designing 7-stranded Ig backbones, we carried out hundreds of independent blueprint-based trajectories folding each target topology in one step followed with a backbone relaxation using strand pairing constraints. We encouraged correct formation of strand pairs using custom python scripts writing distance and angle constraints specifying backbone hydrogen bond pairing at each pair of residue positions. The generated backbones were subsequently filtered based on their match with the secondary structure and ABEGO torsion bins

specified in the corresponding blueprint files, and their long-range backbone hydrogen bond energy (lr_hb_bb score term). We carried out FastDesign[45] calculations using the Rosetta all-atom energy function ref2015[46] to optimize sidechain identities and conformations with low-energy, efficiently packing the protein core, and compatible with their solvent accessibility. Designed sequences were filtered based on the average total energy, Holes score[47], buried hydrophobic surface, and sidechain-backbone hydrogen bond energy (for better stabilizing β-arch geometry). For loop residue positions, we restricted amino acid identities based on sequence profiles derived from naturally occurring loops with the same ABEGO torsion bins[5].

### Sequence-structure compatibility evaluation

The local compatibility between the designed sequences and structures was evaluated based on fragment quality. Sequence-structure pairs were considered locally compatible if for all residue positions at least one of the picked 9-mer fragments (based on sequence and secondary structure similarity with the design) had a RMSD below 1.0 Å. For designs fulfilling this requirement, we assessed their folding by Rosetta ab initio structure prediction in two steps. We started screening hundreds of designs quickly with biased forward folding simulations[5] (BFF) using the three 9- and 3-mers closer in RMSD to the design. Those designs with a substantial fraction (>10%) of BFF trajectories sampling structures with RMSDs to the design below 1.5 Å were then selected for standard Rosetta ab initio structure prediction[25]. We ran AlphaFold[26] and the PyRosetta version of RoseTTAFold[27] with a local installation and using default parameters.

### Docking calculations

HADDOCK[48] was used for the evaluation of the crystallographic interface of the design. We picked the first chain from the dIG8-CC crystal structure and used two copies of this monomer for all two-body docking simulations. Taking advantage of the ability of HADDOCK to build missing atoms, we constructed the mutants by renaming and removing all atoms but those forming the backbone (N, $C_\alpha$, C, O) and the $C_\beta$ (to maintain sidechain directionality). For the simulations targeting the crystallographic interface, we selected all residues pertaining to the first and seventh strands (segments 1–7 and 65–70) as active residues to drive the docking. For the ones aiming to the opposite interface, all residues from the third and fourth strands (segments 30–35 and 39–45) were instead used as active residues. For all docking simulations, we defined two different sets of symmetry restraints as follows: (1) we applied C2 symmetry restraints to assure a 180° symmetry axis between both molecules, and (2) enabled noncrystallographic restraints (NCS) to enforce identical intermolecular contacts. All remaining docking and analysis parameters were kept as default. In terms of analysis, the generated models were evaluated by the default HADDOCK scoring function. This mathematical approximation is a weighted linear combination of different energy terms including: van der Waals and electrostatic intermolecular energies, a desolvation potential and a distance restraint energy term. The scoring step is followed by a clustering procedure based on the fraction of common contacts, and the resulting clusters are re-ranked according to the average HADDOCK score of the best 4 cluster members. For comparison purposes, we used the exact same set of parameters for all docking simulations and selected the top model from the best-ranked cluster.

### Design of disulfide bonds

The identification of the position of disulfide bonds was carried out with a motif hashing protocol[33]. 30,000 examples of native disulfide geometries were extracted from high-resolution protein crystal structures in the PDB. The relative orientation of the backbone atoms was calculated by determining the translation and rotation matrix between the two sets of backbone atoms. These translation and rotation matrices were hashed and stored in a hash table with the associated conformation of the sidechains. Once the hash table has been completed by including all of the examples of disulfides from the PDB, the hash table can be utilized to place disulfides into de novo proteins by evaluating the relative orientation within a designed protein to find which residue pairs match an example from the hash table. All of the code necessary to generate the hash tables and run the disulfide placement protocol can be found in https://github.com/atom-moyer/stapler.

### Design of EF-hand calcium-binding motifs

A minimal EF-hand motif from Protein Data Bank (PDB) accession code 1NKF[49] was generated by truncating the PDB file 3-dimensional coordinates to the minimal $Ca^{2+}$-binding sequence DKDGDGYISAAE. RosettaRemodel[50] blueprint files were generated from the 3-dimensional coordinates of the dIG8 computational model and minimal EF-hand motif, and an in-house script used to write RosettaRemodel blueprint files for domain insertion of the minimal EF-hand motif into dIG8. 132 blueprint files were generated to insert the EF-hand motif after residues 8, 28, and 61 of dIG8 while systematically sampling N-terminal linker lengths of 0–3 residues with β-sheet secondary structure and C-terminal linker lengths of 0–10 residues with α-helical secondary structure. RosettaRemodel was run three times for each blueprint file using the `pyrosetta.distributed` and `dask` python modules[51–53]. Linker compositions were de novo designed in RosettaRemodel using specific sets of amino acids defined in the blueprint files at each position of the N-terminal and C-terminal linkers while preventing repacking of EF-hand motif sidechain rotamers required for chelating $Ca^{2+}$. Out of 396 domain insertion simulations, 86 successfully closed the N-terminal and C-terminal linkers producing single-chain decoys. On each decoy, a custom PyRosetta script was run to append a $Ca^{2+}$ ion into the EF-hand motif. Decoys were then relaxed via Monte Carlo sampling of protein sidechain repacking and protein sidechain and backbone minimization steps with a full-atom Cartesian coordinate energy function[46] with coordinate constraints applied to the aspartate and glutamate residues chelating the $Ca^{2+}$ ion. The 86 resulting designs were scored in RosettaScripts[44] with an in-house XML script. Concomitantly, each of the 86 designs were forward folded[25] after temporarily stripping out the $Ca^{2+}$ ion from each decoy, and the ff_metric algorithm used to evaluate funnels[54]. To select designs for experimental validation, the following computational protein design metric filters were applied: buns_all_heavy_ball ≤ 1.0; buns_all_heavy_ball_interface ≤ 1.0; total_score_res ≤ −3.7; geometry = 1.0. Filtered designs were ranked ascending primarily on buns_all_heavy_ball, ascending secondarily on ff_metric, and ascending tertiarily on total_score_res. To experimentally test designs at the three domain insertion sites, the top three ranked designs at each of the three domain insertion sites were selected. To experimentally test designs with the shortest N-terminal and C-terminal linkers, the top three ranked designs with up to a 3-residue N-terminal linker and up to a 2-residue C-terminal linker were selected. 12 designs in total were selected for experimental characterization after mutating positions compatible with disulfide bonds to cysteines.

### Recombinant expression and purification of the designed proteins for biophysical studies

Synthetic genes encoding for the selected amino acid sequences were ordered from Genscript and cloned into the pET-28b+ expression vector, with the genes of interest inserted within NdeI and XhoI restriction sites and the pET28b backbone encoding an N-terminal, thrombin-cleavable His6-tag. *Escherichia coli* BL21 (DE3) competent cells (Sigma) were transformed with these plasmids, and starter cultures from single colonies were grown overnight at 37 °C in Luria-Bertani (LB) medium supplemented with kanamycin. Overnight cultures were used to inoculate 50 ml of Studier autoinduction media[55] with antibiotic as done in a previous study[56]. Cells were harvested by

centrifugation and resuspended in a 25 mL lysis buffer (20 mM imidazole in PBS containing protease inhibitors), and lysed by microfluidizer. PBS buffer contained 20 mM NaPO4, 150 mM NaCl, pH 7.4. After removal of insoluble pellets, the lysates were loaded onto nickel affinity gravity columns to purify the designed proteins by immobilized metal-affinity chromatography (IMAC). The expression of purified proteins was assessed by SDS-polyacrylamide gel; and protein concentrations were estimated from the absorbance at 280 nm measured on a NanoDrop spectrophotometer (ThermoScientific) with extinction coefficients predicted from the amino acid sequences using the ProtParam tool (https://web.expasy.org/protparam/). Proteins were further purified by size-exclusion chromatography using a Superdex 75 10/300 GL (GE Healthcare) column.

## Circular dichroism

Far-UV circular dichroism measurements were carried out with a JASCO spectrometer. Wavelength scans were measured from 260 to 195 nm at temperatures between 25 and 95 °C with a 1 mm path-length cuvette. Protein samples were prepared in PBS buffer (pH 7.4) at a concentration of 0.3–0.4 mg/mL. GdnCl solutions were prepared by dissolving GdnCl salt into PBS buffer and checking the refractive index.

## Size-exclusion chromatography coupled to multiple-angle light scattering (SEC-MALS)

To ascertain the oligomerisation state of dIG proteins, SEC-MALS was performed in a Dawn Helios II apparatus (Wyatt Technologies) coupled to a SEC Superdex 75 Increase 10/300 column. The column was equilibrated with PBS or buffer B at 25 °C and operated at a flow rate of 0.5 mL/min. A total volume of 100–165 μL of protein solution at 1.0–3.0 mg/mL was employed for each sample. Data processing and analysis proceeded with Astra 7 software (Wyatt Technologies), for which a typical $dn/dc$ value for proteins (0.185 mL/g) was assumed.

## Protein production for crystallization studies

The original thrombin site of plasmids pET28-dIG8-CC and pET28-dIG14 was replaced with a Tobacco-Etch-Virus peptidase (TEV) recognition site via NcoI and Nde employing forward and reverse primers (Eurofins) listed in Supplementary Table 7. The generated plasmids, pET28*-dIG8-CC and pET28*-dIG14, were mixed at 100 mg each in Takara buffer (50 mM Tris-HCl, 10 mM magnesium chloride, 1 mM dithiothreitol, 100 mM sodium chloride, pH 7.5), annealed by slowly cooling down the sample to room temperature following 4 min at 94 °C, and ligated into the doubly digested plasmid. For pET28*-dIG14, the original thrombin-cleavable N-terminal His6-tag was removed and four histidine residues were added to the protein C-terminus by PCR using NcoI and XhoI sites (see Supplementary Table 7 for primers). Of note, due to the cloning strategy, dIG18-CC and dIG-14 proteins were preceded by a G–H–M and a M–G motif, respectively. All PCR reactions and ligations were performed using Phusion High Fidelity DNA polymerase and T4 Ligase, and ligation products were transformed into chemically competent E. coli DH5-α cells for multiplication (all Thermo Fisher Scientific). Plasmids were purified with the E.Z.N.A. Plasmid Mini Kit I (Omega Bio-Tek) and verified by sequencing (Eurofins and Macrogen).

For protein expression, competent E. coli BL21 (DE3) cells (Sigma) were transformed with the pET28*-dIG8-CC and pET28*-dIG14 plasmids and grown on LB plates supplemented with 100 μg/mL kanamycin. Single colonies were selected to inoculate 5-mL starter cultures of this medium and incubated overnight at 37 °C under shaking. Respective 1-mL aliquots were used to inoculate 500 mL of the same medium. Once cultures reached OD600 ≈ 0.6, protein expression was induced with 0.5 mM IPTG (Fisher Bioreagents), and cultures were incubated overnight at 18 °C. Cells were harvested by centrifugation (3500 × $g$, 30 min, 4 °C) and resuspended in cold buffer A (50 mM Tris-HCl, 250 mM sodium chloride, pH 7.5), supplemented with 10 mM

imidazole, EDTA-free cOmplete Protease Inhibitor Cocktail (Roche Life Sciences), and DNase I (Roche Life Sciences). Cells were lysed using a cell disrupter (Constant Systems) operated at 135 MPa, and soluble protein was clarified by centrifugation (50,000 × $g$, 1 h, 4 °C) and subsequently passed through a 0.22-μm filter (Merck Millipore).

For immobilised-metal affinity chromatography (IMAC[57]), proteins were captured on nickel-sepharose HisTrap HP columns (Cytiva), which had previously been washed and pre-equilibrated with buffer A plus either 500 mM or 20 mM imidazole, respectively. Column-bound dIG14 was extensively washed with a gradient of 20-to-150 mM imidazole in buffer A and eluted with a gradient of 200-to-300 mM imidazole in buffer A. Column-bound dIG8-CC was washed and eluted with buffer A containing 20 mM and 300 mM imidazole, respectively.

Fractions containing the dIG8-CC protein were then buffer-exchanged to buffer B (20 mM Tris-HCl, 150 mM sodium chloride, pH 7.5) in a HiPrep 26/10 desalting column (GE Healthcare), and incubated overnight at 4 °C with inhouse-produced His6-tagged TEV peptidase at a peptidase:substrate ratio of 1:20 (w/w) and 1 mM dithiothreitol for fusion-tag removal. After centrifugation (50,000 × $g$, 1 h, 4 °C) and filtration (0.22-μm), the clarified dIG8-CC protein was loaded again onto the HisTrap HP column for reverse IMAC with buffer A plus 20 mM imidazole, which retained tagged protein and TEV, and had untagged dIG8-CC in the flow-through. The bound proteins were eventually eluted with buffer A plus 300 mM imidazole for column regeneration.

Untagged dIG8-CC and dIG14 were polished by size-exclusion chromatography (SEC) with buffer B in a Superdex 75 Increase 10/300 GL column (Cytiva) attached to an ÄKTA Purifier 10 apparatus. Protein purity was assessed by 20% SDS-PAGE stained with Coomassie Brilliant Blue (Sigma). PageRule Unstained Broad Range Protein Ladder and PageRuler Plus Prestained Protein Ladder (both Thermo Fisher Scientific) were used as molecular-mass markers. To concentrate protein samples, ultrafiltration was performed using Vivaspin 15 and Vivaspin 2 Hydrosart devices (Sartorius Stedim Biotech) of 2-kDa molecular-mass cutoff. Protein concentrations were determined either by the BCA Protein Assay Kit (Thermo Fisher Scientific) with bovine serum albumin as a standard or by $A_{280}$ using a BioDrop Duo+ apparatus (Biochrom). Supplementary Fig. 16 provides proof of the effective protein purification procedures.

## Protein crystallization

Crystallization screenings using the sitting-drop vapor diffusion method were performed at the joint IRB/IBMB Automated Crystallography Platform (www.ibmb.csic.es/en/facilities/automated-crystallographic-platform) at Barcelona Science Park (Catalonia, Spain). Screening solutions were prepared and dispensed into the reservoir wells of 96 × 2-well MRC crystallization plates (Innovadyne Technologies) by a Freedom EVO robot (Tecan). These reservoir solutions were employed to pipet crystallization nanodrops of 100 nL each of reservoir and protein solution into the shallow crystallization wells of the plates, which were subsequently incubated in steady-temperature crystal farms (Bruker) at 4 °C or 20 °C.

After refinement of initial hit conditions, suitable dIG14 crystals appeared at 20 °C in drops consisting of 0.5 μL protein solution (at 1.9 mg/mL in buffer B) and 0.5 μL reservoir solution (0.1 M sodium acetate, 0.2 M calcium chloride, 20% w/v polyethylene glycol [PEG] 1500, pH 5.5). Crystals were cryoprotected with reservoir solution supplemented with 20% glycerol, harvested using 0.1–0.2 mm nylon loops (Hampton), and flash-vitrified in liquid nitrogen. The best tetragonal dIG8-CC crystals were obtained at 20 °C in drops containing 0.5 μL protein solution (at 30 mg/mL in buffer B) and 0.5 μL reservoir solution (0.1 M Bis-Tris, 0.2 M calcium chloride, 20% w/v PEG 3350, 10% v/v ethylene glycol, pH 6.5). Crystals were directly harvested using 0.1–0.2 mm loops, and flash-vitrified in liquid nitrogen. Proper orthorhombic dIG8-CC crystals resulted from the same condition as

the tetragonal ones except that magnesium chloride and glycerol replaced calcium chloride and ethylene glycol, respectively. Furthermore, 0.25 mL of 5% n-dodecyl-N,N-dimethylamine-N-oxide (w/v) was included as an additive. These crystals were cryoprotected with reservoir solution supplemented with 20% glycerol, harvested with elliptical 0.02–0.2 mm LithoLoops (Molecular Dimensions), and flash-vitrified in liquid nitrogen.

## Diffraction data collection and structure solution

X-ray diffraction data were recorded at 100 K on a Pilatus 6 M pixel detector (Dectris) at the XALOC beamline[58] of the ALBA synchrotron (Cerdanyola, Catalonia, Spain) and on an EIGER X 4 M detector (Dectris) at the ID30A-3 beamline[59] of the ESRF synchrotron (Grenoble, France). Diffraction data were processed with programs Xds[60] and Xscale, and transformed with Xdsconv to MTZ-format for the Phenix[61] and CCP4[62] suites of programs. Analysis of the data with Xtriage[63] within Phenix and Pointless[64] within CCP4 confirmed the respective space groups and indicated absence of twinning and translational non-crystallographic symmetry. Supplementary Table S5 provides essential statistics on data collection and processing.

The structure of dIG8-CC, both in its tetragonal ($P4_12_12$; 2.30 Å) and orthorhombic ($C222_1$; 2.05 Å) space groups, was solved by molecular replacement with the Phaser[65] program employing the coordinates of the designed structure. The tetragonal crystals contained four protomers (chains A–D) in the asymmetric unit (a.u.) arranged as two dimers, and the calculations gave final refined values of the translation function Z-score (TFZ) and log-likelihood gain (LLG) of 14.5 and 307, respectively. Subsequently, the adequately rotated and translated molecules were subjected to successive rounds of manual model building with the Coot program[66] alternating with crystallographic refinement with the Refine protocol of Phenix[67], which included translation/libration/screw-motion (TLS) refinement and non-crystallographic symmetry (NCS) restraints. The final model included residues $R^1$–$G^{70}$ of each protomer preceded by $M^0$, $H^{-1}$, and, in chain D only, $G^{-2}$ from the upstream linker, as well as 22 solvent molecules. The orthorhombic crystals were solved as the tetragonal ones with final refined TFZ and LLG values of 11.9 and 263, respectively. Model building and refinement proceeded as above. The final model encompassed residues $R^1$–$G^{70}$ of each protomer preceded by $M^0$ and $H^{-1}$, plus one magnesium cation and 34 solvent molecules. Cysteines $C^{21}$ and $C^{60}$ were present in both disulfide-linked and unbound conformations in all protomers of both crystal forms. The occupancy of the disulfide bond in the two crystal structures ranges between 0.00 and 0.67 across the eight protomers, with a mean occupancy of 0.47 and 0.41 in each of the structures (Supplementary Table 6).

The structure of dIG14 in a yet different space group ($P4_32_12$; 2.50 Å) with two molecules per a.u. was likewise solved by molecular replacement, with final refined TFZ and LLG values amounting to 17.4 and 269, respectively. The phases derived from the adequately rotated and translated molecules were subjected to a density modification and automatic model building step under twofold averaging with the Autobuild routine[68] of Phenix, which produced a Fourier map that assisted model building as aforementioned. Crystallographic refinement was also performed as above except that both Phenix and the BUSTER package[69] were employed. The final model comprised $R^1$–$G^{68}$ of protomer A and $R^1$–$F^{74}$ of protomer B, either preceded by $G^0$ and $M^{-1}$ from the upstream linker, as well as 15 solvent molecules.

Supplementary Table 5 provides essential statistics on the final refined models, which were validated through the wwPDB Validation Service at https://validate-rcsb-1.wwpdb.org/validservice and deposited with the PDB at www.pdb.org with accession codes: 7SKN (design: dIG8-CC; space group: $P4_12_12$), 7SKO (design: dIG8-CC; space group: $C222_1$), and 7SKP (design: dIG14; space group: $P4_32_12$). Supplementary

Fig. 17 shows 2Fo-Fc electron density maps for the three protein structures.

## $Tb^{3+}$ binding luminescence measurements

To measure the $Tb^{3+}$ luminescence of samples dIG8-CC and EF61_dIG8-CC (in buffer 20 mM Tris, 50 mM NaCl, pH 7.4), time-resolved luminescence emission spectra and intensities were measured on a Synergy H1 hybrid multi-mode reader (BioTek) in flat bottom, black polystyrene, 96-well half-area microplates (Corning 3694). A stock solution of terbium(III) chloride ($TbCl_3$) (Sigma-Aldrich, 451304-1G) was prepared in the same protein buffer. Time-resolved luminescence intensities were measured using excitation wavelength $\lambda_{ex}$ = 280 nm and emission wavelength $\lambda_{em}$ = 544 nm with a delay of 300 µs, 1 ms collection time, and 100 readings per data point. Time-resolved luminescence emission spectra between 520 nm and 570 nm was collected in 2 nm increments and smoothed with a Savitzky-Golay filter of order 3 (Fig. 5h). For $Tb^{3+}$ titrations, samples were incubated for 3 h and the collected time-resolved luminescence emission intensities at $\lambda_{em}$ = 544 nm were normalized to obtain protein-bound fractions, and the normalized data was fit to the equilibrium binding equation with a Hill coefficient of 1 using non-linear least squares regression (Fig. 5i; Supplementary Fig. 15a). $Ca^{2+}$ binding was measured by titrating $CaCl_2$ prepared in the same protein sample buffer into 20 µM EF61_dIG8-CC and 100 µM $Tb^{3+}$, and measuring the decrease of time-resolved luminescence emission intensity at $\lambda_{em}$ = 544 nm (Supplementary Fig. 15b).

## Protein expression of isotopically labeled proteins for NMR

Plasmids were transformed into BL21 (DE3) expression strain of E. coli (Invitrogen) and grown in 50 mL of Luria Broth containing 50 µg/mL of kanamycin and grown at 37 °C with shaking overnight. After ~18 h, the 50 mL starter culture was used to inoculate 500 mL of minimal labeling media (M9), containing N15 labeled Ammonium Chloride at 50 mM and C13 glucose to 0.25% (w/v), as well as trace metals, 25 mM $Na_2HPO_4$, 25 mM $KH_2PO_4$, and 5 mM $Na_2SO_4$. The culture was returned to 37 °C, at 250 rpm and allowed to reach $OD_{600}$ ~0.7–1.0. To induce expression 1 mM of IPTG was added and the temperature was reduced to 25 °C to allow the culture to express overnight. Cells were harvested by centrifugation at 5000 × g for 20 min then resuspended with 40 mL of Lysis Buffer (20 mM Tris 250 mM NaCl 0.25% Chaps pH 8) and lysed with a Microfluidics M110P Microfluidizer at 18,000 psi. The lysed cells were clarified using centrifugation at 24,000 × g for 30 min. The labeled protein in the soluble fraction was purified using Immobilized Metal Affinity Chromatography (IMAC) using standard methods (Qiagen Ni-NTA resin). The purified protein was then concentrated to 2 mL and purified by FPLC size-exclusion chromatography using a Superdex 75 10/300 GL (GE Healthcare) column into 20 mM $NaPO_4$ 150 mM NaCl pH 7.5. The efficiency of labeling was confirmed using mass spectrometry.

## Nuclear magnetic resonance spectroscopy

NMR data were acquired at 30 °C on Bruker spectrometers operating at 600 or 800 MHz, equipped with cryogenic probes. His-tagged double-labeled ($^{15}$N, $^{13}$C) dIG21 and $^{15}$N-labeled dIG14 constructs were dissolved in PBS buffer (pH 7.5, 150 mM NaCl) at concentrations of ~150–200 µM. For dIG21, triple-resonance backbone spectra, and a 3D NH-NOESY spectrum, were acquired with non-uniform sampling schemes in the indirect dimensions and were reconstructed by the multi-dimensional decomposition software qMDD[70], interfaced with NMRPipe[71]. Peak picking was performed using NMRFAM-SPARKY[72,73], and the automated in-house program FMCGUI, which employs an ABACUS approach, was used to aid in the assignment of backbone resonances[74,75].

## Visualization of protein structures and image rendering

Images of protein structures were created with PyMOL[76].

## Reporting summary

Further information on research design is available in the Nature Research Reporting Summary linked to this article.

## Data availability

The data that support this study are available from the corresponding authors upon request. Coordinates and structure factors have been deposited in the Research Collaboratory for Structural Bioinformatics Protein Data Bank with the accession codes 7SKN (dIG8-CC, tetragonal space group), 7SKO (dIG8-CC, orthorhombic space group) and 7SKP (dIG14). All the designed protein structures experimentally tested are available as Supplementary Data 1, and their corresponding sequences are provided in Supplementary Table 2. Further structural analyses (for loops, cross-β motifs, and Ig designs), biochemical and biophysical characterization of the designs, structure prediction calculations, sequence analysis, and X-ray crystallography statistics are provided as Supplementary Figures and Tables. The AlphaFold Protein Structure database used for structural analysis is freely available (https://alphafold.ebi.ac.uk). Source data are provided with this paper.

## Code availability

The Rosetta macromolecular modeling suite (http://www.rosettacommons.org) is freely available to academic and non-commercial users. Computational protocols used for analyzing and designing protein structures are available at https://github.com/emarcos/immunoglobulin_design.

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

## Acknowledgements

We are grateful to Laura Company and Joan Pous from the joint IBMB/IRB Automated Crystallography Platform and the Protein Purification Service for assistance during SEC-MALS, purification procedures, and crystallization experiments. We thank Lauren Carter and Cameron Chow for assistance with SEC-MALS experiments and NMR sample preparation at the Institute for Protein Design. We also thank Minkyung Baek for assistance with structure predictions with RoseTTAFold. The authors would further like to thank the ESRF and ALBA synchrotrons for beam-time allocation and the respective beamline staff for assistance during diffraction data collection. We acknowledge computing resources provided by Rosetta@Home volunteers, the Galicia Supercomputing Center (CESGA), and the Red Española de Supercomputación (grants BCV-2021-1-0014 and BCV-2021-3-0010). This research was supported by grants from the Spanish Ministry of Science and Innovation (RYC2018-025295-I, EUR2020-112164, and PID2020-120098GA-I00). This study was also supported in part by grants from Spanish and Catalan public and private bodies (grant/fellowship references MCIN/AEI/10.13039/501100011033/PID2019-107725RG-I00, 2017SGR3 and Fundació "La Marató de TV3" 201815). S.R.M. acknowledges grant BES2016-076877 from the Spanish State Agency for Research (MCIN/AEI/10.13039/501100011033) and the European Social Fund "ESF invests in your future". U.E. was funded by a Beatriu de Pinós post-doctoral fellowship (AGAUR-MSCA COFUND 2018BP00163. J.R.T. was supported by an

EMBO postdoctoral fellowship (under grant agreement ALTF 145-2021). J.C.K. was supported by a National Science Foundation Graduate Research Fellowship (grant DGE-1256082). D.B. and T.M.C. acknowledge the Howard Hughes Medical Institute. We thank the Princess Margaret Cancer Centre for funding of the NMR facility. The Structural Genomics Consortium is a registered charity (no: 1097737) that receives funds from Bayer AG, Boehringer Ingelheim, Bristol Myers Squibb, Genentech, Genome Canada through Ontario Genomics Institute [OGI-196], EU/EFPIA/OICR/McGill/KTH/Diamond Innovative Medicines Initiative 2 Joint Undertaking [EUbOPEN grant 875510], Janssen, Merck KGaA (aka EMD in Canada and US), Pfizer and Takeda. The content herein is solely the responsibility of the authors and does not necessarily represent the official views of the funding agencies.

## Author contributions

E.M., T.M.C., F.X.G.R., and D.B. designed the research. T.M.C. carried out design calculations, protein expression, purification, and CD experiments. S.R.M. cloned, expressed, purified, and characterized proteins. S.R.M., T.G., and U.E. crystallized proteins, and U.E. collected and analyzed diffraction data. T.M.C. and J.C.K. designed and experimentally tested EF-hand terbium-binding loops. J.R.T. carried out docking calculations. M.N. expressed, purified, and performed CD and terbium-binding experiments. F.X.G.R. solved crystal structures. H.K.H. analyzed design structural diversity. A.M. provided crosslinking scripts for disulfide bridging. S.H. and C.H.A. carried out NMR spectroscopy. E.M. set up the design methods, carried out design calculations, and performed the structural analyses. E.M., T.M.C., F.X.G.R., and D.B. prepared the manuscript with input from all authors.

## Competing interests

D.B., T.M.C., J.C.K., S.R.M., U.E., F.X.G.R., and E.M. have filed a US provisional patent application 63/316,733 on discoveries described in this manuscript. The other authors declare no competing interests.
