## [Peer Review File · Nature Communications]

REVIEWER COMMENTS

Reviewer #1 (Remarks to the Author):

The paper by Chidyausiku and colleagues presents a computational workflow for the de novo design of Ig-like folds. This challenge has been a long-standing one in computational design, particularly because beta structure only proteins have been rather challenging for computational design. The paper is well written and the results clearly presented. The computational and experimental methodologies and results are sound.

The strategy proposed relies on defining strict structural "rules" based on the frequency of particular structural motifs that are the building blocks of the fold to be designed. This strategy has been used extensively by the Baker group to design many other folds using de novo approaches, however given the challenge of designing such folds this paper reports a very important achievement. The designed sequences were extensively characterized experimentally, both biochemically as well as structurally. Experimentally many of the sequences were found to be dimers rather than monomers, which is not particularly problematic, nevertheless it begs the question: why not having used some type of negative design to avoid edge strand dimerization. It would be worth to include a short paragraph about this aspect.

The structural characterization of one of the designed proteins (dIG14) was somehow disappointing given that some considerable differences were observed in the design, which however could even be related to the crystallization artefacts. In a follow up design (dIG18-CC), where a disulfide was design to stabilize the correct configuration of the fold.

The functional loop scaffolding is an effort to functionalize some the dIG scaffolds with some type of functionality, but besides the fact that it shows that new loop motifs can be added to the scaffold, it does not show particularly striking results.

The computational methods are available, which is valuable addition to the paper.

Specific points:

- I) discuss the lack of a negative design step to avoid dimerization
- II) clarify in figure 4 the disulfide design step which is very unclear
- III) I would suggest to the authors to add the SEC results in the main text
- IV) in table S2 – despite the lack of sequence relationships according to the different sequence search algorithms, for instance the pdbid 2r39 is in fact an Ig like fold – would be important to be clear about this, mentioning that despite the very low sequence identity and distant evolutionary links there are some detectable sequence signatures
- V) for figure 7 – I understand the motivation of the representation but it would be much more informative to plot the maximum TM score of the designs to any native structure or the distribution of the designs vs the native Ig-like folds

Reviewer #2 (Remarks to the Author):

Review of Chidyausiku ... Marcos "De novo design of immunoglobulin-like domains"

This an important, landmark piece of work that will be of great interest and built upon by a spectrum of readers, from theoretical design and folding to focused translational medicine. The research is thoroughly and capably executed. However, the paper omits discussion of several major aspects, is too-briefly and confusingly written and illustrated, and is therefore very hard work to understand. Because of its really important contribution, the needed discussions, clarifications, and figure improvements are very well worth doing, and I look forward to seeing it in a revised form.

Immunoglobulin domains have a number of highly conserved features in the core beta-sandwich which presumably give enough stability to permit enormous variability of sequence and conformation in the hypervariable loops that bind the amazingly wide

variety of antigens. This work concentrates almost exclusively on the two conserved beta-arches which form what they have named the "cross-beta" motif of high-contact-order organization central to the immunoglobulin fold. This is a relatively recent and clearly productive perspective for analyzing these structures. However, the paper makes essentially no mention of other conserved immunoglobulin features. One is the SS bond that is both cross-barrel and between beta-arches, with its contacting Trp (and these designs later add a differently placed SS between beta-arches for stability). Another is the Greek key motif that has long been hypothesized to aid folding by turning the non-local beta-arch contacts into local ones by being part of a long beta hairpin that can curl to make a second pair of strands that look non-local in a topology diagram or by sequence numbers (Fig 100 in Richardson 1981 *Anatomy and Taxonomy of Protein Structure*, *Adv Prot Chem* 34:167). The 3rd strand of the Greek key motif forms the necessary top-to-bottom sequence connection between the two beta arches, and it is present in every one of the final designs shown here. However, the fact that all therefore have a Greek-key motif, and its possible influence on folding and on effective contact order, is not mentioned at all. Such a discussion of the Greek key must be added. More details of the SS in DIG8-CC are needed. The design-model conformation has several near-eclipsed dihedrals, which would make it relatively unstable. Are they also eclipsed for the database entry, and for the crystal structure? What is the mean and range of occupancy for the open vs bonded alternates seen in the many crystal-structure copies? Those open disulfides are presumably some combination of occurrence in solution and radiation damage during data collection (an effect that must be mentioned), although both indicate relatively poor SS stability. What is the specific definition that identifies the end residues of a beta strand? In my experience, many cases are ambiguous. Those identities are central to the design process, and without a specification this work is not reproducible. Natural IGGs have variable loops only at one end of the sandwich (at bottom in all these figures), which is not noted in the paper. It would seem that one advantage of these designs is that a binding site could be designed at either end. However, the edge rather than face dimer here is probably a disadvantage because a long line of 6 loops would probably make it harder to use most of them in a binding site.

Try to help your readers. The multiple supplemental files must have descriptive filenames, and at the end of the main text there must be a clear explanation of what is in the supplement, so readers do not have to download all of it and try to figure out what's in each file and which ones they need. For things like ABEGO, you should explain in a few words before referring readers to details in the Supplement. For figures that compare a model with a crystal structure, the caption needs to specify which PDB code and which copy are shown. At the end of the crystallography section of Methods, pair the 3 PDB codes explicitly with which construct and space group.

The figures are very comprehensive, and have a nice appearance and apparent clarity. However, their diagrammatic conventions and labeling are sometimes inherently confusing and very often inconsistent between and within figures, and Fig 1 of the main text dumps readers into the details without first a clear presentation of the main framework of analysis. Fig 1 S of the Supplement should be in the main text and come first, with two simple visual modifications shown in the attached Fig.1s_mod.png: a strong line along the sequence, and hairpin labels that actually show where the hairpins are. The caption should say "and backbone H-bond patterns (thin lines) between paired b-strands along the sequence". Then it would be an excellent, intuitive introductory figure.

In current Fig. 1, strand numbers for the beta strands start out as s1, s2, s3, s4 in parts a & b and then the same 4 strands are labeled as s2, s3, s5, s6 in the full domain. Don't label/number the initial strands in a & b, just label the two arches. Somewhat harder, but extremely helpful, would be instead of up/down arrows, which suggest the strand relationships, to represent the sidechain direction of strand-end residues as short arrow pairs pointing left or right. Or possibly "I" or "In" for inward-pointing and "X" or "Ex" for outward-pointing, so Ex-Ex, In-Ex, ... In the caption for part c, say "folding simulations (gray shaded squares)" and "Black-outline boxes highlight". I'm puzzled

how there can be only 3 combinations seen in natural Ig domains, since beta-arch 1 is a hypervariable loop.

Jane Richardson

Reviewer #3 (Remarks to the Author):

In the present manuscript, the authors approach de novo design of immunoglobulin (Ig) domains by addressing an important feature of the Ig fold architecture, the non-local cross-beta structure connecting two beta-sheets. The structures of designed domains have been confirmed using X-ray crystallography and they also provide evidence of their functionality as scaffolds for functional loops.

The manuscript is very clearly written and a pleasure to read as the reasoning behind the conceptualization and the experiments is straight forward. To that, the images are very clear, well presented and support the understanding of the inference.

In detail, Rosetta folding simulations were first used to generate different versions of 7-stranded scaffolds featuring a multitude of diverse pairs of cross-beta motifs, including beta-arch loops and beta-arch helices, and developed a set of rules by which the formation and structure of these motifs can be designed to be compatible with a set of beta strands. Using these principles, seven Ig-like topologies were designed de novo, avoiding cysteines and minimizing risk of edge-to-edge orientations. Folding of the novel designs was inspected using folding simulations and most promising designs were processed with ab initio folding simulation starting from an extended chain, as well as AlphaFold and RoseTTAFold.

31 designs were then selected for experimental characterization, of which 24 could be expressed in the soluble form in *E. coli*, which is a great success. Moreover, 8 of those were monodisperse, of near-all beta-sheet structure as judged by far-UV circular dichroism and were mostly extremely thermostable, with T_m over 95°C. Of these 8 designs, five were dimeric, one monomeric, for which also a well-dispersed NMR spectrum was determined, and one was in equilibrium between these states. Also chemical stability with unfolding in the increasing concentration of GdnCl was inspected. The most stable design, which was still folded in 5 M GdnCl, was found to be dimeric and crystallization was performed to discover an edge-to-edge dimer was formed. The AlphaFold monomer prediction recapitulated the design model, however the AF multimer correctly predicted the monomer subunits in the crystal structure.

One of the mutants was additionally stabilized with a disulphide bond between the beta strands, which enabled its crystallization. In contrast with the previous example, a 14-strand beta-sandwich was formed with an edge-to-edge interface between the N- and C-terminal beta-strands. Into this mutant, an EF-hand calcium binding motif was grafted; 12 designs were experimentally tested and one with graft in C-terminal loops was well folded, monodisperse and Tb³⁺ luminescence could be induced by FRET. Tb³⁺ binding was shown to be specific using Ca²⁺ displacement.

Alltogether, the manuscript shows an elegant and very efficient method of de novo design of Ig-like domains with high stability and good biochemical properties, and highlights the importance of cross-beta motifs for design. As such it is very valuable to the scientific community. My only comment would be that the Discussion section is very short. The authors argue that edge-to-edge dimers that mostly result from the designed proteins are favorable because they shield the edges which are responsible for aggregation propensity. What options are there for designing face-to-face dimer forming oligomers (such as in naturally occurring Igs, e.g. VL and VH domains)? They also offer an interesting option of designing single chain – connected dimers based on described scaffolds, similar to scFvs, but their molecules are not yet binders. In what way will the introduction of antigen binding site(s) impact the stability and the dimerization propensity? What extent of freedom in design of possible binding sites do the authors envision, and what could they look like? I also believe that this would be of significance

to make the last statement of the abstract, mentioning "antibody-like scaffolds" stronger.

Changes made in response to the comments of the Reviewers

In the following sections, the comments of the Reviewers are written in italics. Our responses to the each of the items are presented under their comments in blue font. Excerpts from the text are written in smaller blue font and indented.

Reviewer 1:

The paper by Chidyausiku and colleagues presents a computational workflow for the de novo design of Ig-like folds. This challenge has been a long-standing one in computational design, particularly because beta structure only proteins have been rather challenging for computational design. The paper is well written and the results clearly presented. The computational and experimental methodologies and results are sound.

The strategy proposed relies on defining strict structural “rules” based on the frequency of particular structural motifs that are the building blocks of the fold to be designed. This strategy has been used extensively by the Baker group to design many other folds using de novo approaches, however given the challenge of designing such folds this paper reports a very important achievement. The designed sequences were extensively characterized experimentally, both biochemically as well as structurally.

We thank Reviewer 1 for these positive comments.

Experimentally many of the sequences were found to be dimers rather than monomers, which is not particularly problematic, nevertheless it begs the question: why not having used some type of negative design to avoid edge strand dimerization. It would be worth to include a short paragraph about this aspect.

A new paragraph has been added to the Discussion section as described below in our response to “Specific Point I”.

The structural characterization of one of the designed proteins (dIG14) was somehow disappointing given that some considerable differences were observed in the design, which however could even be related to the crystallization artefacts. In a follow up design (dIG18-CC), where a disulfide was design to stabilize the correct configuration of the fold.

The functional loop scaffolding is an effort to functionalize some the dIG scaffolds with some type of functionality, but besides the fact that it shows that new loop motifs can be added to the scaffold, it does not show particularly striking results.

The computational methods are available, which is valuable addition to the paper.

Specific points:

I) discuss the lack of a negative design step to avoid dimerization

We acknowledge that we have not included an explicit negative design step, which involves enumerating the conformational states (all of the possible homodimer interfaces in this case) that need to be destabilized during the sequence optimization process. We have added a new paragraph to the Discussion section describing possible negative design strategies and, in particular, the challenge of explicitly enumerating negative states to be destabilized in the context of this work:

Several of the designs tended to dimerize in solution, highlighting design challenges in preventing self-interactions between β -sheets. Solvent-exposed β -strand edges favor intermolecular β -strand pairing through backbone hydrogen bonds (between the unpaired NH- and CO- groups) and hydrophobic interactions at the interface between monomers. As in previous *de novo* β -sheet design studies (5, 7, 8), we used an implicit negative design strategy to disfavor association by favoring polar or charged amino acids at inward-facing positions of the edge β -strands to weaken interface sidechain interactions. Explicit negative design against possible edge-to-edge dimer interfaces is an alternative, but remains challenging as it requires enumerating many possible negative states: the crystal structures of two designs show two possible interfaces (one including structural rearrangement of the monomer), and we cannot rule out the possibility that other dimer interfaces formed in designs that were not crystallized (via parallel or antiparallel edge-strand pairing with varied register shifts). Alternatively, negative design against edge-to-edge interfaces can be encoded in protein backbone irregularities – e.g. β -bulges, prolines or short protective β -strands – disfavoring the ideal geometry for hydrogen-bonded β -strand pairing (41).

However, we did use an implicit negative design approach that was found to be useful in previous *de novo* protein design studies, which involves favoring polar or charged amino acids at inward-pointing residue positions of the edge β -strands. In the description of our sequence design approach in the main text we have now added a sentence explicitly mentioning such implicit negative design:

As an implicit negative design strategy against edge-to-edge interactions promoting aggregation, we incorporated at least one inward-facing polar or charged amino acid (TQKRE) (24) into each solvent-exposed edge β -strand.

II) clarify in figure 4 the disulfide design step which is very unclear

Disulfide positions were computationally designed as described in the Methods section, and now in the main text we have added a sentence summarizing the design approach:

As disulfide bonds with high sequence separation are more stabilizing due to greater unfolded state entropy reduction, we computationally designed disulfide bonds between β -strands not forming a β -hairpin using a hash-based disulfide placement protocol (33) which searches for transformations between pairs of residue positions compatible with naturally occurring disulfide bond geometries (see Methods).

To better show where the disulfide bond was designed, we have also added a new panel (Fig. 5a) showing the design model with the disulfide bond highlighted.

III) I would suggest to the authors to add the SEC results in the main text

We have now added the SEC results for the dIG14 and dIG8-CC designs in Figures 4a and 5b, respectively, together with a caption specifying the theoretical molecular weight of the monomers as a reference.

IV) in table S2 – despite the lack of sequence relationships according to the different sequence search algorithms, for instance the pdbid 2r39 is in fact an Ig like fold – would be important to be clear about this, mentioning that despite the very low sequence identity and distant evolutionary links there are some detectable sequence signatures

Certainly, the high secondary structure propensity of our designs together with loops favoring either β -hairpin or β -arch connections enabled the sequence search algorithms to find local sequence-similarity matches with a natural Ig domain in some cases. In a new paragraph of the Discussion section describing differences between the designed and natural Ig domains, we have added the following sentence:

The designs contain cross- β motifs less twisted than those from natural Ig domains, and their overall structural (average TM-score of 0.54) and sequence (Supplementary Table 2) similarity is very low (HHPred did identify matches to short segments of β -sandwiches, including one Ig domain (PDB accession code 2R39), with locally similar alternating patterns of hydrophobic and polar amino acids typical of β -strands).

V) for figure 7 – I understand the motivation of the representation but it would be much more informative to plot the maximum TM score of the designs to any native structure or the distribution of the designs vs the native Ig-like folds

We agree with the Reviewer that this can be a very clear way for conveying a similar message. Supplementary Fig. 7 is now Supplementary Fig. 6 (due to a figure reorganization). We have added a new figure panel (Supplementary Fig. 6b) showing the distribution of TM-scores between all designs and native Ig domains, with an average TM-score of 0.54 (indicating low structural similarity but in the same fold) and

a standard deviation of 0.06. In the main text, when describing structural differences between designs and natural Ig domains, we have also added a sentence:

The designs also differ substantially from natural Ig domains in global structure (with an average \pm s.d. TM-score (31) of 0.54 ± 0.06 ; Supplementary Fig. 6), and cross- β twist rotation (close to zero, which are infrequent in natural Ig domains; Supplementary Table 3).

We thank Reviewer 1 for their helpful suggestions and comments.

Reviewer 2:

Review of Chidyausiku ... Marcos "De novo design of immunoglobulin-like domains"

This an important, landmark piece of work that will be of great interest and built upon by a spectrum of readers, from theoretical design and folding to focused translational medicine. The research is thoroughly and capably executed. However, the paper omits discussion of several major aspects, is too-briefly and confusingly written and illustrated, and is therefore very hard work to understand. Because of its really important contribution, the needed discussions, clarifications, and figure improvements are very well worth doing, and I look forward to seeing it in a revised form.

We thank Dr. Jane Richardson for her positive comments. We have taken into consideration all of her suggestions for improving the clarity of the manuscript (please see below).

Immunoglobulin domains have a number of highly conserved features in the core beta-sandwich which presumably give enough stability to permit enormous variability of sequence and conformation in the hypervariable loops that bind the amazingly wide variety of antigens. This work concentrates almost exclusively on the two conserved beta-arches which form what they have named the "cross-beta" motif of high-contact-order organization central to the immunoglobulin fold. This is a relatively recent and clearly productive perspective for analyzing these structures. However, the paper makes essentially no mention of other conserved immunoglobulin features.

One is the SS bond that is both cross-barrel and between beta-arches, with its contacting Trp (and these designs later add a differently placed SS between beta-arches for stability). Another is the Greek key motif that has long been hypothesized to aid folding by turning the non-local beta-arch contacts into local ones by being part of a long beta hairpin that can curl to make a second pair of strands that look non-local in a topology diagram or by sequence numbers (Fig 100 in Richardson 1981 Anatomy and Taxonomy of Protein Structure, Adv Prot Chem 34:167). The 3rd strand of the Greek key motif forms the necessary top-to-bottom sequence connection between the two beta arches, and it is present in every one of the final designs shown here. However, the fact that all therefore have a Greek-key motif, and its possible influence on folding and on effective contact order, is not mentioned at all. Such a discussion of the Greek key must be added.

We agree with the Reviewer that we focused our description of the Ig fold solely on its overall topology and the core cross- β motif because these were the main features we considered for design. We acknowledge that it will valuable to add a description of

additional conserved features that are well-described in the literature and how they are linked to the design approach we have followed. Both in the main text and the Discussion section we have extended this description and added clarifications. For example, we now clarified that the cross- β motif that we described results from two Greek key motifs that, as locally folding units, mediate the formation of non-local contacts. To help clarify this concept, in the new Figure 1 (as suggested below by the Reviewer) we have explicitly showed the two Greek key motifs of the Ig fold and how they build the cross- β motif. In addition, we have also added a reference to other common features such as the sheet-to-sheet disulfide, the buried tryptophan in strand B, and the tyrosine corner.

We have made the following addition to the second paragraph of the main text:

The four constituent cross- β strands (S_2 , S_3 , S_5 , S_6) correspond to the B, C, E and F β -strands that build the common structural core of Ig domains found in nature (10, 11), and for which some sequence signatures related to stability or function have been reported – e.g. a disulfide bridge between the B and F β -strands, a buried tryptophan in β -strand B (11, 14) or the tyrosine corner (15) between β -strand C and the loop connecting β -strands E and F. The non-local cross- β structure comprises two Greek key super-secondary structures (16, 17) involving four consecutive β -strands in which the first is paired to the last (Fig. 1b).

To the second paragraph of the Discussion section, we have added:

The cross- β motifs of our designs differ from natural ones in several ways. Our cross- β motifs are formed by combining short β -arch loops not seen in natural Ig domains (Fig. 2c), which generally have more complex loops (including a complementarity-determining region (CDR) in the first β -arch of the cross- β motif found in antigen-binding regions of antibodies), and are stabilized by hydrophobic interactions without incorporating sequence motifs typically found in the core strands B, C, E and F of natural Ig domains. For example, the disulfide bond of DIG8-CC is between two β -strands paired in the same β -sheet in contrast to the sheet-to-sheet disulfide bridge found between strands B and F in many Ig domains. The tyrosine corner which stabilizes Greek keys in many natural β -barrels and β -sandwiches (15, 18) was also not needed in our designs. These differences in sequence requirements reflect the substantial structural differences between our designs and natural Ig domains.

More details of the SS in DIG8-CC are needed. The design-model conformation has several near-eclipsed dihedrals, which would make it relatively unstable. Are they also eclipsed for the database entry, and for the crystal structure? What is the mean and range of occupancy for the open vs bonded alternates seen in the many crystal-structure copies? Those open disulfides are presumably some combination of occurrence in

solution and radiation damage during data collection (an effect that must be mentioned), although both indicate relatively poor SS stability.

The disulfide bond was designed by searching for placements of native disulfide bond geometries in the backbone of dIG8 as described in the Methods section, and now further mentioned in the main text:

As disulfide bonds with high sequence separation are more stabilizing due to greater unfolded state entropy reduction, we computationally designed disulfide bonds between β -strands not forming a β -hairpin using a hash-based disulfide placement protocol (33) which searches for transformations between pairs of residue positions compatible with naturally occurring disulfide bond geometries (see Methods).

It is certainly possible that a range of disulfide bond stabilities could be obtained with this computational protocol. We have added a new Supplementary Fig. 12 showing distributions for the five dihedral angles from the database of native disulfide bonds, and compared them with the values of the designed disulfide bridge in dIG8-CC. We acknowledge that two out of the five designed dihedrals have low frequencies in the database, and we hypothesize that this could be related to the relatively low disulfide bond stability that was observed based on the low occupancies in the crystal structures. It is worth noting that the disulfide bonds observed with partial occupancies in the crystal structures have the same geometry as in the design. We have also added Supplementary Table 6 with the occupancies of the disulfide bonds (see below).

We have added the following sentences to the Results section:

The sidechain of residue C21 was found in two different conformations, disulfide-bonded with C60 as in the design and unbound (Supplementary Table 6), which suggests low stability of the disulfide bond (Supplementary Fig. 12) and that it is not essential for proper folding of dIG8-CC. This is consistent with the high stability determined for parental dIG8 without the disulfide bridge (Supplementary Fig. 7).

The Supplementary Fig. 12 caption gives details about the analysis on disulfide bond dihedrals:

Supplementary Fig. 12. Dihedrals of the designed dIG8-CC disulfide bond in comparison with natural distributions. a, Five dihedrals describing the geometry of the designed disulfide bond (spheres and sticks) between C21 and C60. b, Distribution of χ_1 (and χ_1') dihedral angles obtained from a database of ~30,000 native disulfide bond geometries that was used for design (see Methods). The corresponding dihedral angles of the dIG8-CC design are represented as dashed vertical lines ($\chi_1 = -60.3^\circ$ in blue and $\chi_1' = -59.9^\circ$ in green). c, Distribution of χ_2 (and χ_2') dihedral angles obtained from the database of native

disulfide bond geometries. The corresponding dihedral angles of the dIG8-CC design are represented as dashed vertical lines ($\chi_2 = -130.6^\circ$ in blue and $\chi_2' = -72.0^\circ$ in green). d, Distribution of the χ_3 dihedral angle obtained from the database of native disulfide bond geometries. The corresponding dihedral angle of the dIG8-CC design is represented as dashed vertical lines ($\chi_3 = 117.1^\circ$ in blue). Two of the five disulfide dihedral angles (χ_2 and χ_3) are not frequently observed in distributions from naturally occurring disulfides, which is likely associated with the low disulfide bond stability suggested by the crystal structures.

The occupancy of the disulfide bond in the two crystal structures ranges between 0.00 and 0.67 across the eight protomers, with a mean occupancy of 0.47 and 0.41 in each of the structures. We acknowledge that radiation damage can contribute to decrease disulfide bond occupancies, and also the His-tag removal step, which was performed in the presence of a reducing agent (1 mM dithiothreitol) for optimal cleavage. We have added Supplementary Table 6 (see caption below) with the occupancies of the disulfide bonds in each protomer of the crystal structures, and a comment on the possible factors contributing to such partial disulfide bond formation:

Supplementary Table 6. Occupancies for the bound and unbound disulfides between C21 and C60 observed in the crystal structures of dIG8-CC. Neither rotamer nor Ramachandran outliers were identified for the cysteines across the eight protomers. Despite possible radiation damage and partial reduction from the hexahistidine tag removal step with TEV-protease (see Methods), the close to 50% formation suggests low disulfide bond stability.

We have also included a sentence in the Crystallography methods section describing the mean and range of occupancies of the disulfide bonds in the crystal structure:

Cysteines C21 and C60 were present in both disulfide-linked and unbound conformations in all protomers of both crystal forms. The occupancy of the disulfide bond in the two crystal structures ranges between 0.00 and 0.67 across the eight protomers, with a mean occupancy of 0.47 and 0.41 in each of the structures (Supplementary Table 6).

What is the specific definition that identifies the end residues of a beta strand? In my experience, many cases are ambiguous. Those identities are central to the design process, and without a specification this work is not reproducible.

In the Methods section, we have specified that the secondary structures are assigned with the DSSP algorithm. Based on this secondary structure definition, we take the first and last residue of each β -strand as the end residues. This has been common practice in many previous *de novo* protein design studies, although we know there are alternative

approaches for secondary structure assignment and defining end residues of β -strands. To further clarify this idea we have added the following sentence to the Methods section:

(the first and last residue of each assigned β -strand were considered the end residues connecting to the loops).

Natural IGGs have variable loops only at one end of the sandwich (at bottom in all these figures), which is not noted in the paper. It would seem that one advantage of these designs is that a binding site could be designed at either end. However, the edge rather than face dimer here is probably a disadvantage because a long line of 6 loops would probably make it harder to use most of them in a binding site.

We thank the Reviewer for making this observation. We acknowledge that edge-to-edge and face-to-face dimers may have pros and cons, and future designs and experiments on scaffolding multiple loops into such arrangements will contribute to identify proper ways of using *de novo* designed Ig dimers for targeted binding. The relative orientation and distances between loop sites from different chains differs in these two dimeric arrangements, and hence we expect that, depending on the target structure and the loops involved, one of two arrangements will be better suited than the other for designing shape-complementary binding interfaces. We have now extended the paragraph of the Discussion section about the edge-to-edge dimers:

The edge-to-edge dimer interfaces in the crystal structures of our designs differ from those found between the heavy- and light-chains of antibodies, which are arranged face-to-face. For engineering antibody-like formats presenting several loops targeting one or multiple epitopes, designing dimeric Ig interfaces through the β -sandwich edge formed by the terminal β -strands has the advantage over face-to-face dimers of decreasing the number of exposed β -strand edges, thereby reducing aggregation-propensity. It will likely be useful to custom-design both edge-to-edge and face-to-face dimers from our *de novo* Ig domains; these would present loops from the two monomers in different relative orientations, and depending on the target structure and the loops involved, one of these two arrangements will likely be better suited than the other for designing shape-complementary binding interfaces. Another advantage of controlling the orientation of dimer interfaces is that the N- and C-termini of the two monomeric subunits can be positioned in close proximity to allow fusion through short or compact connections into rigid and hyperstable single-chain constructs –similar in spirit to single-chain variable fragments (scFvs) but with greater structural control and higher stability.

Based on the Ig orientation we have used in our figures, the equivalent positions for the CDRs in antibodies are located at bottom, as in the EF-hand motif we inserted into dIG8-CC. In the Discussion section, we have added a sentence commenting on this and

also the possibility that other positions located at the top could be in principle functionalized in the *de novo* designed Ig domains:

In antibodies, the CDRs are located on one side of the β -sandwich (at the bottom given the orientation displayed in Figs. 1-5), and we inserted the terbium binding motif on this side, but the robustness of our scaffolds could allow insertions on the other side as well.

Try to help your readers. The multiple supplemental files must have descriptive filenames, and at the end of the main text there must be a clear explanation of what is in the supplement, so readers do not have to download all of it and try to figure out what's in each file and which ones they need. For things like ABEGO, you should explain in a few words before referring readers to details in the Supplement. For figures that compare a model with a crystal structure, the caption needs to specify which PDB code and which copy are shown. At the end of the crystallography section of Methods, pair the 3 PDB codes explicitly with which construct and space group.

For additional clarity, we have now extended the title of those supplementary figures that were too short. We have referenced all Supplementary Figures and Tables in the main text, and at the end we have now summarized all the data that is found in the Supplementary Materials:

Further structural analyses (for loops, cross- β motifs and Ig designs), biochemical and biophysical characterization of the designs, structure prediction calculations, sequence analysis and X-ray crystallography statistics are provided as Supplementary Figures and Tables.

In the main text, we have now included a definition of ABEGO:

It is convenient to describe the backbone geometry of loop residue positions with ABEGO torsion bins representing different areas of the Ramachandran plot (“A”, right-handed α -helix region; “B”, extended region; “E”, extended region with positive ϕ ; “G”, left-handed α -helix region; and “O”, if the peptide bond deviates from planarity) (see Supplementary Fig. 1a for a definition).

In Figures 4 and 5, we have now added the specification on the PDB accession codes of the crystal structures and the chain used for comparison with the design. We have also made an addition to the crystallography section as suggested by the Reviewer:

Supplementary Table 4 provides essential statistics on the final refined models, which were validated through the wwPDB Validation Service at <https://validate.rcsb-1.wwpdb.org/validservice> and deposited with the PDB at www.pdb.org with accession

codes: 7SKN (design: dIG8-CC; space group: P41212), 7SKO (design: dIG8-CC; space group: C2221), and 7SKP (design: dIG14; space group: P43212).

The figures are very comprehensive, and have a nice appearance and apparent clarity. However, their diagrammatic conventions and labeling are sometimes inherently confusing and very often inconsistent between and within figures, and Fig 1 of the main text dumps readers into the details without first a clear presentation of the main framework of analysis. Fig 1 S of the Supplement should be in the main text and come first, with two simple visual modifications shown in the attached Fig.1s_mod.png: a strong line along the sequence, and hairpin labels that actually show where the hairpins are. The caption should say “and backbone H-bond patterns (thin lines) between paired β -strands along the sequence”. Then it would be an excellent, intuitive introductory figure.

We thank the suggestion made by the Reviewer and now turned the Supplementary Fig. 1 into the new Fig. 1 of the main text with further modifications for improved clarity. The rest of figures have been renumbered properly along with their references in the text. As described above, in such new Figure 1, we have included a description of the Greek key motifs present in the Ig fold and how they are related to the formation of the central cross- β motif.

In current Fig. 1, strand numbers for the beta strands start out as s1, s2, s3, s4 in parts a & b and then the same 4 strands are labeled as s2, s3, s5, s6 in the full domain. Don't label/number the initial strands in a & b, just label the two arches. Somewhat harder, but extremely helpful, would be instead of up/down arrows, which suggest the strand relationships, to represent the sidechain direction of strand-end residues as short arrow pairs pointing left or right. Or possibly “I” or “In” for inward-pointing and “X” or “Ex” for outward-pointing, so Ex-Ex, In-Ex, ... In the caption for part c, say “folding simulations (gray shaded squares)” and “Black-outline boxes highlight”.

We thank the Reviewer for the suggestion for better clarity and incorporated these changes to the Figure (now Figure 2). We adopted the convention of “In” for inward-pointing and “Out” for outward-pointing. We have also adapted the Supplementary Figures to this clearer naming convention.

I'm puzzled how there can be only 3 combinations seen in natural Ig domains, since beta-arch 1 is a hypervariable loop.

It is worth clarifying that the analysis made in Fig. 2c was done with 20 β -arch loops (i.e., 5 for each of the 4 possible sidechain directions of adjacent residues) frequently

observed in naturally occurring protein structures (not only Ig domains), and contained a maximum of 5 residues. The analysis was done in this way because we were interested in short, structured loops, but most natural Ig domains are built with longer β -arch loops – including CDRs, as in the case of antibodies and nanobodies. If we would extend the analysis to more loops for each of the 4 types, we would expect to find more combinations observed in natural Ig domains, but this would become computationally very intensive due to the large number of combinations. We did not extend the analysis to more loops because this was already sufficient for the purpose of our design approach. We have added two sentences describing this observation:

In the Results section describing Figure 2:

Of the short β -arch loops we considered for design, only a few are present in the cross- β motifs of naturally occurring Ig domains (Fig. 2c), which are mostly built by longer or hypervariable loops (as is the case of the first β -arch).

And in a paragraph of the Discussion section about differences between the designed and natural Ig domains:

The cross- β motifs of our designs differ from natural ones in several ways. Our cross- β motifs are formed by combining short β -arch loops not seen in natural Ig domains (Fig. 2c), which generally have more complex loops (including a complementarity-determining region (CDR) in the first β -arch of the cross- β motif found in antigen-binding regions of antibodies), and are stabilized by hydrophobic interactions without incorporating sequence motifs typically found in the core strands B, C, E and F of natural Ig domains.

We thank Dr. Jane Richardson for her helpful suggestions and comments.

Reviewer 3:

In the present manuscript, the authors approach de novo design of immunoglobulin (Ig) domains by addressing an important feature of the Ig fold architecture, the non-local cross-beta structure connecting two beta-sheets. The structures of designed domains have been confirmed using X-ray crystallography and they also provide evidence of their functionality as scaffolds for functional loops.

The manuscript is very clearly written and a pleasure to read as the reasoning behind the conceptualization and the experiments is straight forward. To that, the images are very clear, well presented and support the understanding of the inference.

In detail, Rosetta folding simulations were first used to generate different versions of 7-stranded scaffolds featuring a multitude of diverse pairs of cross-beta motifs, including beta-arch loops and beta-arch helices, and developed a set of rules by which the formation and structure of these motifs can be designed to be compatible with a set of beta strands. Using these principles, seven Ig-like topologies were designed de novo, avoiding cysteines and minimizing risk of edge-to-edge orientations. Folding of the novel designs was inspected using folding simulations and most promising designs were processed with ab initio folding simulation starting from an extended chain, as well as AlphaFold and RoseTTAFold.

31 designs were then selected for experimental characterization, of which 24 could be expressed in the soluble form in E. coli, which is a great success. Moreover, 8 of those were monodisperse, of near-all beta-sheet structure as judged by far-UV circular dichroism and were mostly extremely thermostable, with T_m over 95°C. Of these 8 designs, five were dimeric, one monomeric, for which also a well-dispersed NMR spectrum was determined, and one was in equilibrium between these states. Also chemical stability with unfolding in the increasing concentration of GdnCl was inspected.

The most stable design, which was still folded in 5 M GdnCl, was found to be dimeric and crystallization was performed to discover an edge-to-edge dimer was formed. The AlphaFold monomer prediction recapitulated the design model, however the AF multimer correctly predicted the monomer subunits in the crystal structure.

One of the mutants was additionally stabilized with a disulphide bond between the beta strands, which enabled its crystallization. In contrast with the previous example, a 14-strand beta-sandwich was formed with an edge-to-edge interface between the N- and C-terminal beta-strands. Into this mutant, an EF-hand calcium binding motif was grafted; 12 designs were experimentally tested and one with graft in C-terminal loops was well folded, monodisperse and Tb^{3+} luminescence could be induced by FRET. Tb^{3+} binding was shown to be specific using Ca^{2+} displacement.

Alltogether, the manuscript shows an elegant and very efficient method of de novo design of Ig-like domains with high stability and good biochemical properties, and highlights the importance of cross-beta motifs for design. As such it is very valuable to the scientific community. My only comment would be that the Discussion section is very short.

We thank the Reviewer for the positive comments. We have now added further discussion of the results to the Discussion section.

The authors argue that edge-to-edge dimers that mostly result from the designed proteins are favorable because they shield the edges which are responsible for aggregation propensity. What options are there for designing face-to-face dimer forming oligomers (such as in naturally occurring Igs, e.g. VL and VH domains)?

While not explored in this study, we suggest that face-to-face dimers could be engineered starting from the monodisperse, monomeric designs, using state-of-the-art computational docking and interface design methodologies – as we did in a previous study where we designed dimers formed via face-to-face β -sheet interfaces [E. Marcos et al., *Principles for designing proteins with cavities formed by curved β sheets*. Science. 355, 201–206 (2017)] (reference #5 in the main text). Such face-to-face dimers would be a natural extension of this study and exciting future direction in the field for completely *de novo* designed antibody-like architectures. We have extended a Discussion paragraph as follows:

The edge-to-edge dimer interfaces in the crystal structures of our designs differ from those found between the heavy- and light-chains of antibodies, which are arranged face-to-face. For engineering antibody-like formats presenting several loops targeting one or multiple epitopes, designing dimeric Ig interfaces through the β -sandwich edge formed by the terminal β -strands has the advantage over face-to-face dimers of decreasing the number of exposed β -strand edges, thereby reducing aggregation-propensity. It will likely be useful to custom-design both edge-to-edge and face-to-face dimers from our *de novo* Ig domains; these would present loops from the two monomers in different relative orientations, and depending on the target structure and the loops involved, one of these two arrangements will likely be better suited than the other for designing shape-complementary binding interfaces.

They also offer an interesting option of designing single chain – connected dimers based on described scaffolds, similar to scFvs, but their molecules are not yet binders. In what way will the introduction of antigen binding site(s) impact the stability and the dimerization propensity?

The single-chain dimers are expected to have higher thermostability and therefore should have increased tolerance for incorporating functional loops. The high stability of edge-to-edge dimer interfaces, as observed for dIG14, suggests that this specific arrangement can be particularly promising for multiple functionalized loops in the single-chain context, and that will be worth exploring in subsequent studies. Such single-chain dimers are expected to be monomeric since the area that is dimerization-prone is already buried in the single-chain and the remaining edge strands do not favor dimerization as noted in the crystal structure of the designs. The Discussion section commenting on this reads now as follows:

Another advantage of controlling the orientation of dimer interfaces is that the N- and C-termini of the two monomeric subunits can be positioned in close proximity to allow fusion through short or compact connections into rigid and hyperstable single-chain constructs –similar in spirit to single-chain variable fragments (scFvs) but with greater structural control and higher stability.

What extent of freedom in design of possible binding sites do the authors envision, and what could they look like? I also believe that this would be of significance to make the last statement of the abstract, mentioning “antibody-like scaffolds” stronger.

We suggest that the computational protocol can be used for incorporating other functionalized loops, such as ligand- and protein-binding motifs, by designing linkers ensuring compatibility between the ends of the loop and the selected anchor points of the designed Ig domains. We have extended the last paragraph of the Discussion section to further comment on this:

The high stability of our designs opens up exciting possibilities for grafting functional loops, as shown for the EF-hand terbium-binding motif inserted into the C-terminal β -hairpin of dIG8-CC. The β -hairpins in our scaffolds can be readily extended to incorporate ligand- and protein-binding motifs, functional peptide motifs, or complementarity-determining regions (CDRs) of antibodies or nanobodies (it is likely more straightforward to insert functional loops into β -hairpins than into β -arches, since the latter tend to form more slowly and need to be highly structured, but this remains to be studied and may vary depending on the loop to be inserted). In antibodies, the CDRs are located on one side of the β -sandwich (at the bottom given the orientation displayed in Figs. 1-5), and we inserted the terbium binding motif on this side, but the robustness of our scaffolds could allow insertions on the other side as well. Ultimately, achieving the structural control over the Ig backbone together with the high expression levels and stability of de novo designed proteins in general should lead to a versatile generation of antibody-like scaffolds with improved properties.

We thank Reviewer 3 for their helpful suggestions and comments.

REVIEWERS' COMMENTS

Reviewer #1 (Remarks to the Author):

The manuscript has improved substantially and it is now in an acceptable form for publication. I thank the authors for all the work they have invested.

Reviewer #2 (Remarks to the Author):

The changes look good to me. I approve, with one very minor change: please add the word "each" at the beginning of the yellow line 6 up from the bottom of page 3.
Jane

Reviewer #3 (Remarks to the Author):

In the revised version, the authors have addressed all points raised and included their comments into the discussion section, which I believe further clarified their views of the achievements and enlightened future prospects of the design protocols used. They have also substantially improved the figures. Both actions contributed to an important improvement of the manuscript, which I think will now attract broader audience. I am happy to recommend the manuscript for publication.

Comments of the Reviewers

Reviewer 1:

The manuscript has improved substantially and it is now in an acceptable form for publication. I thank the authors for all the work they have invested.

We thank Reviewer 1 for these positive comments.

Reviewer 2:

The changes look good to me. I approve, with one very minor change: please add the word "each" at the beginning of the yellow line 6 up from the bottom of page 3.

We thank Reviewer 2 for these positive comments, and added the word “each” as suggested for improved clarity.

Reviewer 3:

In the revised version, the authors have addressed all points raised and included their comments into the discussion section, which I believe further clarified their views of the achievements and enlightened future prospects of the design protocols used. They have also substantially improved the figures. Both actions contributed to an important improvement of the manuscript, which I think will now attract broader audience. I am happy to recommend the manuscript for publication.

We thank Reviewer 3 for these positive comments.